# Lhx2 is a progenitor-intrinsic modulator of Sonic Hedgehog signaling during early retinal neurogenesis

Xiaodong Li[1†], Patrick J Gordon[2†], John A Gaynes[2‡], Alexandra W Fuller[3], Randy Ringuette[4], Clayton P Santiago[5], Valerie Wallace[6], Seth Blackshaw[5], Pulin Li[7], Edward M Levine[1,2,3]*

[1]Vanderbilt Eye Institute, Vanderbilt University Medical Center, Nashville, United States; [2]John A. Moran Eye Center, University of Utah, Salt Lake City, United States; [3]Department of Cell and Developmental Biology, Vanderbilt University, Nashville, United States; [4]Cellular and Molecular Medicine, University of Ottawa, Ottawa, Canada; [5]Solomon H. Snyder Department of Neuroscience, Johns Hopkins University School of Medicine, Baltimore, United States; [6]Donald K. Johnson Eye Institute, Krembil Research Institute, University Health Network, Toronto, Canada; [7]Whitehead Institute of Biomedical Research, Department of Biology, Massachusetts Institute of Technology, Cambridge, United States

*For correspondence:
ed.levine@vumc.org

Present address: †Taconic Biosciences, New York, United States; ‡Department of Physiology and Biophysics, University of Colorado, Colorado, United States

Competing interest: The authors declare that no competing interests exist.

**Abstract** An important question in organogenesis is how tissue-specific transcription factors interact with signaling pathways. In some cases, transcription factors define the context for how signaling pathways elicit tissue- or cell-specific responses, and in others, they influence signaling through transcriptional regulation of signaling components or accessory factors. We previously showed that during optic vesicle patterning, the Lim-homeodomain transcription factor Lhx2 has a contextual role by linking the Sonic Hedgehog (Shh) pathway to downstream targets without regulating the pathway itself. Here, we show that during early retinal neurogenesis in mice, Lhx2 is a multi-level regulator of Shh signaling. Specifically, Lhx2 acts cell autonomously to control the expression of pathway genes required for efficient activation and maintenance of signaling in retinal progenitor cells. The Shh co-receptors Cdon and Gas1 are candidate direct targets of Lhx2 that mediate pathway activation, whereas Lhx2 directly or indirectly promotes the expression of other pathway components important for activation and sustained signaling. We also provide genetic evidence suggesting that Lhx2 has a contextual role by linking the Shh pathway to downstream targets. Through these interactions, Lhx2 establishes the competence for Shh signaling in retinal progenitors and the context for the pathway to promote early retinal neurogenesis. The temporally distinct interactions between Lhx2 and the Shh pathway in retinal development illustrate how transcription factors and signaling pathways adapt to meet stage-dependent requirements of tissue formation.

## Editor's evaluation

This study dissects the complex Shh pathway to explain the phenotypic similarity between Lhx2 and Shh retinal knock-out mice. The authors use multiple converging experimental strategies to show Lhx2 activates the Shh pathway, mainly by up-regulating co-receptors Gas1 and Cdon in retinal progenitor cells. The experiments are creative, and the findings provide evidence that Lhx2 acts in a contextual manner and integrates signalling pathways, conferring enhanced Retinal Progenitor Cells with the competence to respond to Shh. The study provides novel and interesting views on retinal development.

## Introduction

The Sonic Hedgehog (Shh) signaling pathway is essential for the patterning, growth, and histogenesis of multiple tissues. Deregulation resulting in hyperactivation drives tumor growth and hypoactivation leads to congenital brain malformations including holoprosencephaly (*Hong and Krauss, 2018*; *Scales and de Sauvage, 2009*). The canonical pathway is composed of core and accessory components, which contribute to Shh production, availability, reception, intracellular signaling, and transcriptional regulation of target genes (reviewed in *Briscoe and Thérond, 2013*; *Kong et al., 2019*; *Ramsbottom and Pownall, 2016*). At its simplest, Shh signaling occurs when secreted Shh binds to its cognate Patched receptor, relieving inhibition of the Frizzled class GPCR transmembrane protein Smoothened (Smo) (*Figure 1A*). In turn, an intracellular cascade blocks the proteolytic processing of the GLI zinc-finger transcription factors, Gli2 and Gli3, converting them from transcriptional repressors to activators with Gli3 the predominant repressor and Gli2 the predominant activator (*Lipinski et al., 2006*). The net result is the expression of downstream genes which includes the third mammalian GLI paralog Gli1, which like Gli2, functions as a transcriptional activator. Gli1 contributes to feedback regulation after signaling is initiated with primary transcriptional targets being itself (positive feedback) and three negative feedback regulators; *Patched 1* (*Ptch1*)*, Patched 2* (*Ptch2*), and *Hedgehog Interacting Protein* (*Hhip*). These feedbacks contribute to steady state signaling (*Lai et al., 2004*; *Li et al., 2018*), and allow the expression levels of *Gli1*, *Ptch1*, *Ptch2*, and *Hhip* to be used as readouts of signaling.

The ability or *competence* of cells to signal is essential for the pathway to function at the right time, place and magnitude, and many of the core and accessory pathway components contribute to these properties (*Kiecker et al., 2016*). For a responding cell to be competent, positive transducers must be expressed prior to signaling (e.g. Smo, Gli2), but kept in the 'off' state by negative regulators such as Patched, Suppressor of Fused (Sufu), Gli3, and accessory factors such as Protein Kinase A (Pka), Glycogen Synthase Kinase 3 beta (Gsk3b), and Casein Kinase 1 (Ck1). Pathway activation also requires binding of Shh to one of three co-receptors: the integral membrane proteins Cell Adhesion, Oncogene Regulated (Cdon), Brother of Cdon (Boc), or the GPI-anchored protein Growth Arrest Specific 1 (Gas1) (*Allen et al., 2011*; *Izzi et al., 2011*). Thus, pathway regulation is complex even before signaling is initiated. Since many pathway genes are not expressed ubiquitously, understanding how they are regulated in specific contexts can reveal how signaling is tailored to meet the demands of developing tissues and provide insights into developmental timing mechanisms.

In early eye development, Shh signaling is initially required for the regionalization and ventral patterning of the optic neuroepithelium including in the nascent neural retinal domain (*Chiang et al., 1996*; *Gallardo and Bovolenta, 2018*; *Hernández-Bejarano et al., 2015*; *Take-uchi et al., 2003*; *Wang et al., 2015*; *Zhao et al., 2010*). In the newly formed retina, a second interval of signaling occurs in retinal progenitor cells (RPCs) at the onset of neurogenesis, propagating as a central-to-peripheral wave that is coupled to retinal ganglion cell (RGC) production and RGC-derived *Shh* expression (reviewed in *Wallace, 2008*). This coupling is maintained even when RGC production is delayed (*Sigulinsky et al., 2008*) indicating that Shh availability sets the developmental timing of pathway activation because RPCs are competent to signal prior to Shh exposure. How this competence is established in RPCs at the start of retinal neurogenesis has not been addressed.

The LIM-homeodomain transcription factor *Lim-homeobox 2* (*Lhx2*) is a multifunctional regulator of retinal development. Initially expressed in the eye field, Lhx2 expression persists throughout retinal development in RPCs, becoming restricted to Muller glia and a subset of amacrine cells (*de Melo et al., 2012*; *Gordon et al., 2013*; *Tétreault et al., 2009*; *Viczian et al., 2006*; *Yun et al., 2009*). Initially, *Lhx2* is required for optic vesicle patterning and regionalization, lens specification, and optic cup morphogenesis (*Hägglund et al., 2011*; *Porter et al., 1997*; *Roy et al., 2013*; *Seth et al., 2006*; *Tétreault et al., 2009*; *Yun et al., 2009*; *Zuber et al., 2003*). *Lhx2* directs these processes through cell autonomous and nonautonomous mechanisms, in part through regulation of optic vesicle-derived expression of the Bone Morphogenetic Proteins *Bmp4* and *Bmp7* (*Yun et al., 2009*). After optic cup formation, *Lhx2* nonautonomously directs lens development in part through regulation of retinal-derived expression of Fibroblast Growth Factors (FGFs), and possibly *Bmp4* (*Thein et al., 2016* ). Toward the end of retinal histogenesis, *Lhx2* is required at multiple steps in the formation of Muller glia, the sole RPC-derived glial cell type (*de Melo et al., 2016a*; *de Melo et al., 2016b*). In this case, an interaction with Notch signaling is partly responsible, through *Lhx2*-dependent expression

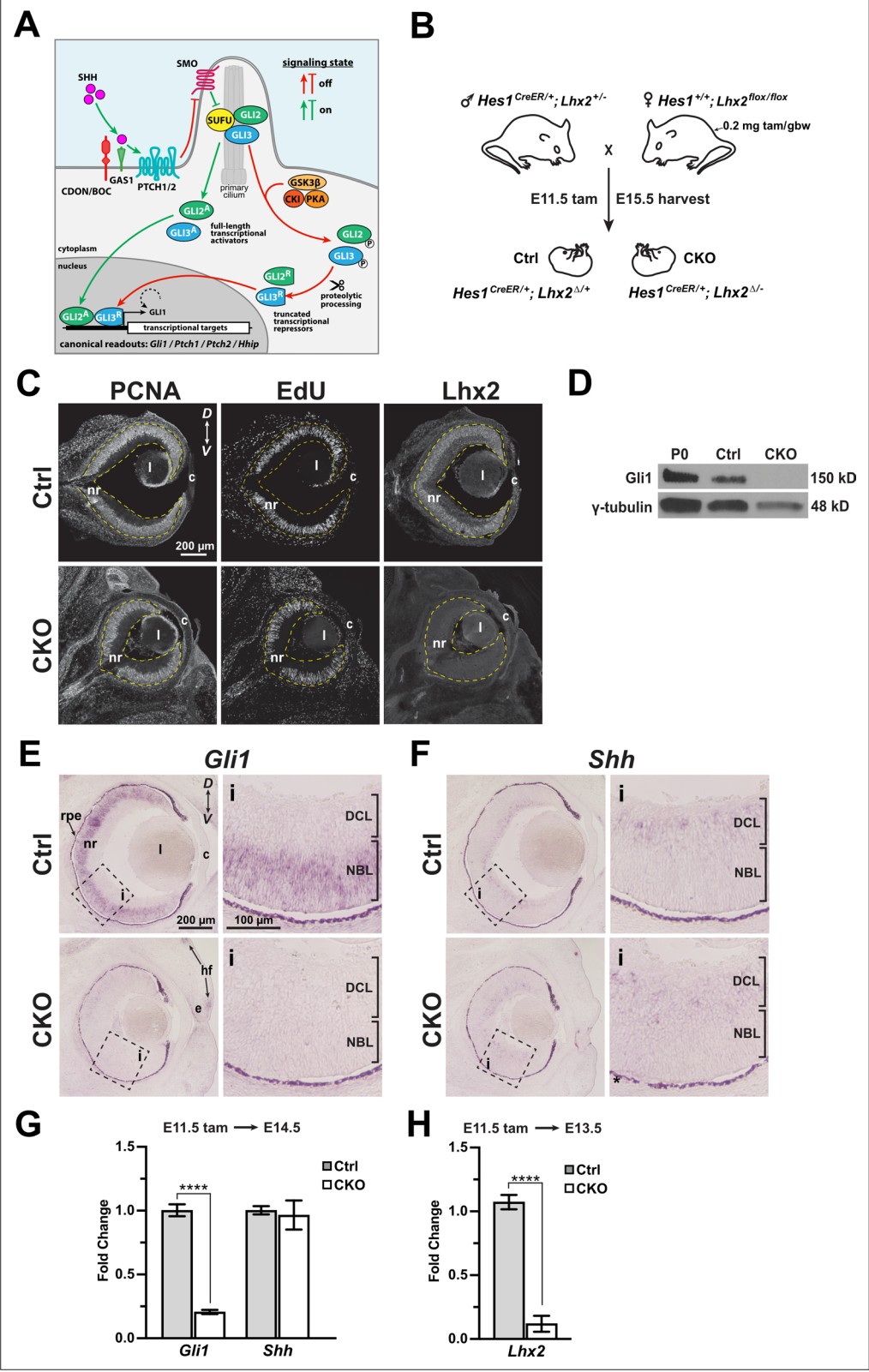

**Figure 1.** *Lhx2* is required for Gli1 expression in RPCs. (**A**) Overview of Hh signaling. See text for details and *Briscoe and Thérond, 2013*; *Kong et al., 2019*; *Ramsbottom and Pownall, 2016* for more comprehensive pathway illustrations and descriptions. (**B**) Genetics and tamoxifen treatment paradigm. Female breeders are also homozygous for *Rosa^{ai14/ai14}* and all embryos are *Rosa^{ai14/+}*, allowing rapid screening for recombined embryos

*Figure 1 continued on next page*

*Figure 1 continued*

with tdTomato expression. Recombined *flox* alleles are indicated by Δ. (**C**) Immunohistology for markers of RPC proliferation (PCNA, EdU) and Lhx2 expression in the E15.5 control and CKO eyes after tamoxifen treatment at E11.5. The retina is contained within the yellow dashed lines. (**D**) Western blot for Gli1 protein expression in P0 wild type, E15.5 Ctrl and CKO retinas following tamoxifen treatment at E11.5. γ-Tubulin served as an internal loading control. (**E, F**) in situ hybridizations for *Gli1* expression in E15.5 Ctrl eyes following tamoxifen treatment at E11.5. (**F**) In situ hybridizations for *Shh* expression in E15.5 Ctrl and CKO eyes following tamoxifen treatment at E11.5. Dashed boxes reveal locations of close-up images (**i**). Dark appearance of the rpe is due to natural pigmentation and does not indicate gene expression. (**G**) qPCR for *Gli1* and *Shh* expression in Ctrl and CKO retinas at E14.5 following tamoxifen at E11.5 (mean+/-S.E.M.; n=4 (Ctrl: *Gli1*); n=6 (CKO: *Gli1*); n=6 (Ctrl: *Shh*); n=8 (CKO: *Shh*); ****, $p_{adj}$=5x10$^{-6}$; unpaired t-tests with multiple comparisons correction; see ***Supplementary file 3*** for statistics). (**H**) qPCR for *Lhx2* expression in Ctrl and CKO retinas at E13.5 following tamoxifen at E11.5 (mean+/-S.E.M.; n=4 (Ctrl); n=4 (CKO); ****, p<0.0001; unpaired t-test; see ***Supplementary file 3*** for statistics). Abbreviations: *D*, dorsal; *V*, ventral; nr, neural retina; rpe, retinal pigment epithelium; l, lens; c, cornea; e, eyelid; hf, hair follicle; DCL, differentiated cell layer; NBL, neuroblast layer.

The online version of this article includes the following source data and figure supplement(s) for figure 1:

**Source data 1.** Western blot probed with Gli1 antibody and reprobed with gamma-Tubulin antibody.

**Figure supplement 1.** Immunohistology for Lhx2 CKO retina at E15.5 following tamoxifen treatment at E11.5.

of ligands (*Dll1*, *Dll3*), the *Notch1* receptor, and downstream transcriptional effectors (*Hes1*, *Hes5*) (***de Melo et al., 2016b***). Thus, the multifunctional nature of *Lhx2* is defined in part by its interactions with developmental signaling pathways, a feature found in other locations of the embryo including the limb where *Lhx2* (and *Lhx9*) are part of the relay connecting FGF signaling from apical ectodermal ridge (AER) and Shh signaling from the zone of polarizing activity (ZPA) (***Tzchori et al., 2009***; ***Watson et al., 2018***), and the hair follicle stem cell compartment where *Lhx2* links Nuclear Factor-Kappa B (NF-kB) signaling to Transforming Growth Factor beta 2 (TGFb2) signaling (***Tomann et al., 2016***). The multifunctional nature of *Lhx2* extends beyond signaling, however, and into multiple developmental processes (i.e. specification, patterning, stem cell/progenitor maintenance, differentiation) for multiple tissues and organ systems, a feature that is perhaps best attributed to its contextual and dynamic interactions with chromatin (***Chou and Tole, 2019***; ***Monahan et al., 2019***; ***Ypsilanti et al., 2021***; ***Zibetti et al., 2019***). Thus, studying *Lhx2* function in specific contexts has provided insights into how a single factor can act as a platform for promoting the execution of complex and varied developmental processes.

We previously reported that RPC-directed *Lhx2* inactivation during embryonic stages of retinal neurogenesis in mice led to reduced proliferation, a failure to maintain the progenitor pool, and altered fated precursor cell production, most notably, the overproduction of RGCs (***Gordon et al., 2013***). These phenotypic features are similar to what occurs when Shh signaling is inhibited at the same developmental stage in the mouse retina or comparable stage in the chick retina (***Wang et al., 2005***; ***Zhang and Yang, 2001***), which prompted us to investigate a potential connection between Lhx2 and Shh signaling.

## Results

### Lhx2 is required for Gli1, but not Shh, expression at the start of retinal neurogenesis

Gli1 is functionally dispensable in mice, but its expression depends on Shh signaling in many tissues including the retina (***Bai et al., 2002***; ***Furimsky and Wallace, 2006***; ***Marigo et al., 1996***; ***McNeill et al., 2012***; ***Park et al., 2000***; ***Sigulinsky et al., 2008***; ***Wang et al., 2002***). Because of its well-established role as a readout of pathway activity, we first asked if Gli1 expression was altered by the loss of Lhx2 function. To bypass the early requirements of Shh signaling in ventral optic vesicle patterning and Lhx2 in optic vesicle regionalization and optic cup morphogenesis (***Fuhrmann, 2010***), *Lhx2* was inactivated in RPCs using the *Hes1*$^{CreER}$ driver by administering tamoxifen to timed pregnant dams at E11.5 (***Figure 1B***). This timing coincides with the onset of retinal neurogenesis and activation of Shh signaling in RPCs. Furthermore, we previously showed that tamoxifen treatment at E11.5 caused microphthalmia and retinal dysplasia that was accompanied by a widespread

reduction in RPCs by E18.5 (*Gordon et al., 2013*). Since *Gli1* is expressed in RPCs, we chose earlier timepoints to examine its expression. At E15.5, eye size is reduced in the CKO (*Figure 1—figure supplement 1A*), but proliferating RPCs were still abundant as indicated by PCNA expression and EdU incorporation, and importantly, Lhx2 was no longer detected (*Figure 1C*). A decrease in the RPC marker Hes1 expression was observed, but DAPI staining and the expression patterns of the RPC marker Cyclin D1, RGC marker Pou4f2, and photoreceptor precursor marker Otx2, were all similar between control and CKO retinas, indicating that tissue organization was not yet severely disrupted (*Figure 1—figure supplement 1B*). Western blot analysis showed that Gli1 protein was markedly reduced, and in-situ hybridization revealed *Gli1* mRNA expression was reduced across the retina (*Figure 1E*). The reduction in *Gli1* expression is unlikely to be due to reduced *Shh* expression since *Shh* was expressed in the CKO in a similar manner as the Ctrl retina (*Figure 1F*). Given the low detectability of Shh and the qualitative nature of the in situ hybridization method, semi-quantitative RT-PCR (qPCR) for *Gli1* and *Shh* was done at E14.5. Consistent with the in-situ expression patterns, *Gli1* expression was reduced and *Shh* was largely unchanged in the CKO retina (*Figure 1G*). Reduced *Lhx2* mRNA expression at E13.5 confirmed that the reduction in *Gli1* was *Lhx2*-dependent (*Figure 1H*).

## Lhx2 inactivation alters the expression of multiple Shh pathway components

The data above do not distinguish between direct regulation of *Gli1* by *Lhx2* or an indirect mechanism where *Lhx2* regulates Shh pathway function. If the latter, we predicted that additional Shh pathway genes would also be altered in the CKO retina. Bulk RNA sequencing (RNA-seq) was done on retinas isolated from E15.5 embryos following tamoxifen treatment at E11.5 (*Figure 2A*). Approximately 12,300 genome mapped features (GMFs; i.e., protein coding genes, pseudogenes, lncRNAs) were identified and examined for differential expression using DESeq2 (*Figure 2B*; *Supplementary file 1*). Overall, 2210 differentially expressed features remained after applying a false discovery rate (FDR) cutoff of 0.001, with 2161 identified as protein coding genes, the next largest category being lncRNAs at 20, and 21 features with rare MGI biotypes such as several types of pseudogene variants. Of these 2210 features, collectively referred to as differentially expressed genes (DEGs), 1184 were downregulated in the CKO within the log2 transformed fold change (log2FC) interval from –5.69 to –0.19, and 1026 DEGs were upregulated within the log2FC interval from 0.17 to 7.69. These statistics are in line with Lhx2 being an essential developmental transcription factor.

qPCR was used to validate the RNA-seq data. In addition to *Lhx2*, the RPC expressed genes *Vsx2*, *Ascl1*, and *Sox2* were downregulated whereas *Pax6*, *Lin28b* and *Prtg* remained unchanged in the RNA-seq data (*Figure 2C*). By qPCR, all genes showed similar trends in relative expression except for *Lhx2*, which showed a more pronounced reduction by qPCR (*Figure 2D*; *Supplementary file 3*). Because the qPCR primers and resulting amplicon are in the third exon, which is deleted in the CKO allele, the discrepancy between the RNA-seq and qPCR is consistent with persistent expression of the mutant transcript rather than incomplete recombination (*Figure 2E*). Overall, the qPCR data align well with the RNA-seq data, and the applied FDR cutoff serves as a reliable indicator of differential expression.

Visualization of the DESeq2 data by volcano plot revealed that none of the Shh pathway genes were among the most divergently expressed using strict cutoffs of log2FC >2.7 and FDR <0.001 (*Figure 2F*). However, *Gli1* was the 8th ranked DEG by FDR and was the top ranked DEG in the Shh pathway. Based on these criteria, no additional Shh pathway genes emerged as candidate targets of *Lhx2* regulation,suggesting that Lhx2 could be regulating *Gli1* independently of Shh signaling or that Lhx2 was exerting more subtle effects on Shh pathway gene expression that culminated in reduced Shh signaling. To assess this possibility in an unbiased manner, gene set overrepresentation analysis and activity state prediction (ASP) analyses were done for the 2210 DEGs using the Canonical Pathways tool (CP) in the Ingenuity Pathway Analysis package (IPA; see Materials and methods). 12 pathways were identified after applying cutoffs of 1.3 for significant overrepresentation and ASP scores of + 2.5 or –2.5 for activated or inhibited signaling, respectively (*Figure 3A*; highlighted in *Supplementary file 2*). In general, activated pathways were associated with neuronal differentiation (orange bars) and inhibited pathways with cell cycle progression (blue bars), which is consistent with the previously reported CKO phenotype (*Gordon et al., 2013*). Shh signaling was identified as an inhibited

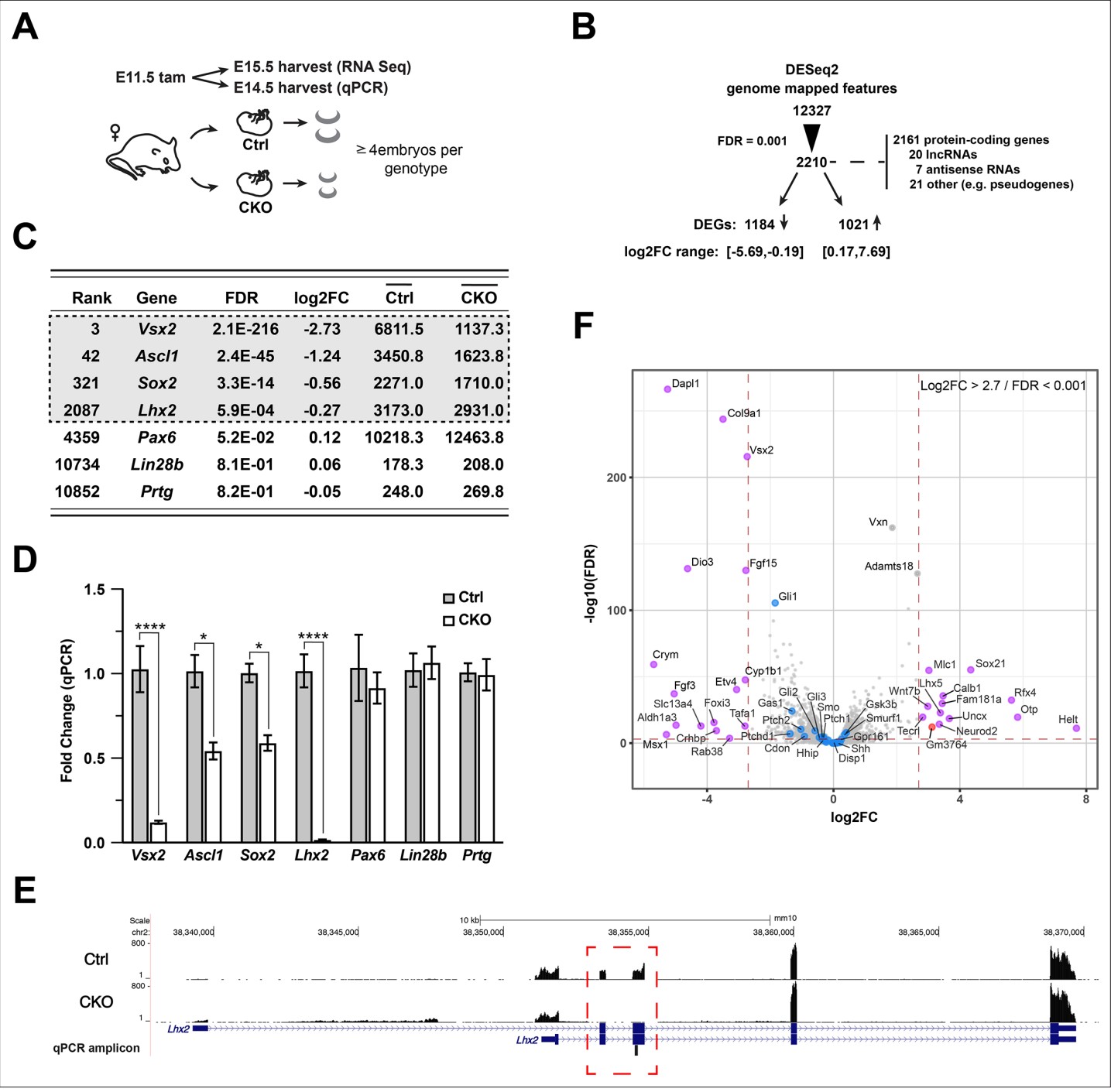

**Figure 2.** Gene expression changes due to *Lhx2* inactivation. (**A**) Schematic of experimental design for RNA sequencing and qPCR. (**B**) Summary of DESeq2 analysis of RNA sequencing datasets. (**C**) DESeq2-derived statistics for progenitor genes with requirements during early retinal neurogenesis. Genes in gray box were within the 0.001 FDR cutoff for differential expression. Averaged counts per gene are shown in the last two columns. (**D**) Relative expression for progenitor genes at E14.5 by qPCR as a function of the fold change from the mean of control for each gene. Only significant comparisons are noted (mean+/-S.E.M.; *, $p_{adj}$<0.05; ****, $p_{adj}$ <0.0001; n=4 (Ctrl: *Vsx2, Ascl1, Lhx2*); n=3 (Ctrl: *Sox2, Pax6*), n=6 (Ctrl: *Lin28b, Prtg*); n=6 (CKO: *Vsx2, Ascl1, Lhx2*); n=4 (CKO: *Sox2, Pax6*); n=8 (CKO: *Lin28b, Prtg*); unpaired t-tests with multiple comparisons correction; see ***Supplementary file 3*** for statistics). (**E**) Coverage plot showing the mutant transcript is expressed and detected by RNA sequencing, indicating that nonsense mediated decay of mutant transcript is not occurring. The lack of reads in exons 2 and 3 (red box) of the CKO reveal the high degree of conditional deletion. Plots are the mean of counts for all samples. The qPCR amplicon located in the deleted exon 3, making the mutant transcript undetectable by qPCR with the Taqman probe used in the study. (**F**) Volcano plot showing selected Shh pathway genes (blue dots) relative to other DEGs. The most divergent DEGs gated on FDR (0.001) and absolute log2FC (2.7) cutoffs (red dashed lines) are highlighted (purple: mRNAs; red: lncRNA; gray: filtered out GMFs).

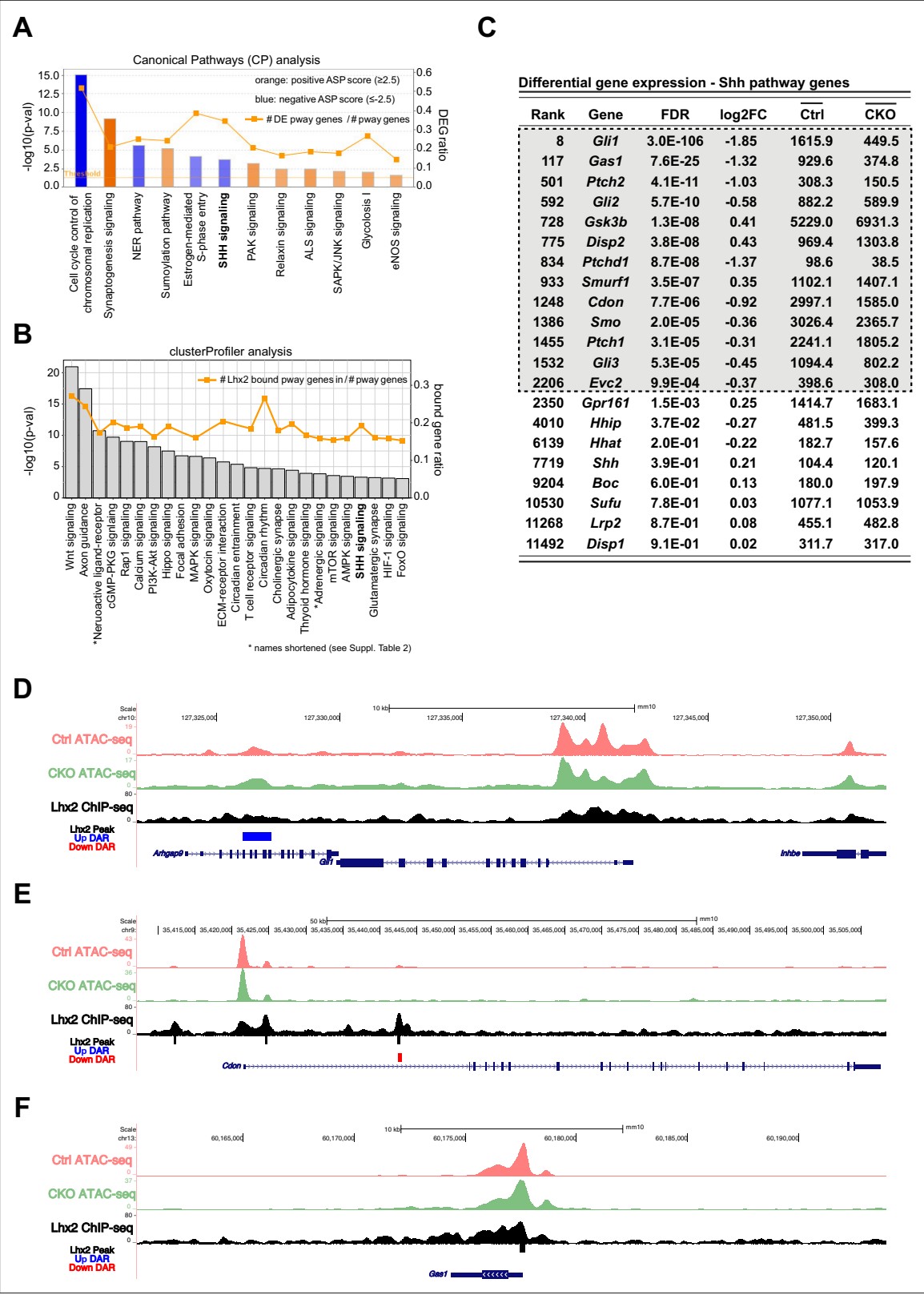

**Figure 3.** Lhx2 is required for the expression of multiple Hedgehog pathway genes. (**A**) Canonical Pathways (CP) analysis for DEGs with an FDR of 0.001 or smaller. Pathways that surpassed the significance cutoff of 1.3 (x-axis) and have an absolute ASP score of 2.5 or higher are shown. Orange bars predict pathway activation and blue bars predict pathway inhibition. Color intensity is directly correlated to ASP score (scores are listed in ***Supplementary file 2***). The line indicates the number of DEGs found in each pathway as a fraction of pathway genes (DEG ratio; right y-axis). (**B**) KEGG pathways associated

*Figure 3 continued on next page*

*Figure 3 continued*

with Lhx2 ChIP-seq peak distribution from E14.5 mouse RPCs using ClusterProfiler. The line indicates the number of genes associated with Lhx2 chromatin binding in each pathway as a fraction of pathway genes (bound gene ratio; right y-axis). (C) Differential gene expression values for canonical genes in the Shh pathway. Genes in the gray box passed the cutoff for DEG designation. (D–F) ATAC-seq and Lhx2 ChIP-seq genomic DNA tracks at *Gli1* (D), *Cdon* (E), and *Gas1* (F) loci from E14.5 RPCs. DARs identified in *Lhx2* CKO RPCs are indicated by red and blue bars. Sites of Lhx2 chromatin binding are indicated by black bars from the ChIP-seq data.

The online version of this article includes the following figure supplement(s) for figure 3:

**Figure supplement 1.** ATAC-seq and Lhx2 ChIP-seq tracks for selected Hh pathway genes within the FDR cutoff for differential expression (0.001).

**Figure supplement 2.** ATAC-seq and Lhx2 ChIP-seq tracks for selected Hh pathway genes outside the FDR cutoff for differential expression (0.001).

pathway (*Figure 3A*) with approximately 35% of the Shh pathway genes identified in the analyzed DEG set (orange line, right Y-axis).

To identify enriched pathways associated with Lhx2 chromatin binding, an Lhx2 ChIP-seq dataset from E14.5 wild type RPCs was analyzed with clusterProfiler against the Kyoto Encyclopedia of Genes and Genomes (KEGG) (*Yu et al., 2012*; *Zibetti et al., 2019*; see Materials and methods). The Shh pathway (listed in KEGG as 'Hedgehog signaling') was identified as overrepresented (*Figure 3B*; *Supplementary file 2*) and was the only pathway identified by both CP and clusterProfiler with the cutoffs applied (compare *Figure 3A and B*). The *Wnt/β -catenin* and *axon guidance* pathways had stronger p-value scores than Shh signaling in both analyses, but their ASP scores failed to reach the cutoff (*Supplementary file 2*). This analysis does not exclude these or other pathways as candidates for *Lhx2* regulation, but the data does support a functional link between *Lhx2* and the Shh pathway that extends beyond direct regulation of *Gli1* by Lhx2.

We next considered possible cause and effect relationships between the differentially expressed Shh pathway genes and signaling state. Multiple pathway genes qualified as DEGs (*Figure 3C*, gray box), and all but *Dispatched 2* (*Disp2*) function at the level of the responding cell. At the receptor level, *Cdon*, *Gas1*, *Ptch1*, *Ptch2*, and *Smo* were reduced in the CKO. Lower expression of *Gas1*, *Cdon*, and *Smo* could negatively impact signaling, consistent with the overall reduction in Shh signaling in the CKO retina. On the other hand, reduced *Ptch1* and *Ptch2* expression should promote signaling, but since this was not the case, their decreased expression levels were likely due to reduced signaling, consistent with their expression levels serving as readouts of pathway activity.

At the intracellular level of the pathway, *Gli2* expression was reduced in the CKO, but *Gli3*, a likely direct target of *Lhx2* (*Zibetti et al., 2019*) was also reduced, potentially offsetting the drop in *Gli2* (*Furimsky and Wallace, 2006*). *Gli2* expression is not typically considered to be dependent on Shh signaling, but its expression was reduced by RPC-specific *Smo* inactivation (*Sakagami et al., 2009*). Of the two upregulated DEGs, only *Gsk3b* is predicted to inhibit signaling by promoting the processing of Gli3 protein into the repressor isoform (*Figure 1A*). Its inhibitory function, however, is normally overridden by ligand-based activation (*Briscoe and Thérond, 2013*). Thus, *Gsk3b* activity could contribute to reduced signaling in the CKO, but only if the pathway is disrupted upstream.

These observations suggest that the changes in expression of *Gas1*, *Cdon*, *Gli3*, and *Gsk3b* in the CKO retina are not dependent on reduced Shh signaling whereas the changes in *Ptch1* and *Ptch2* expression are likely to be due to reduced Shh signaling. It remains possible, however, that the reduced levels of *Gli1*, *Gli2*, and *Smo* in the CKO could be direct outcomes of *Lhx2* inactivation, reduced Shh signaling, or a combination of the two. We reasoned that direct regulation by *Lhx2* on target genes could be inferred from its chromatin binding profile and by changes in chromatin accessibility in *Lhx2* CKO RPCs. To assess this, E14.5 RPC *Lhx2* ChIP-seq data were integrated with ATAC-seq data from E14.5 control and *Lhx2* CKO RPCs (see Methods) (*Zibetti et al., 2019*). Lhx2 binding was not associated with *Gli1*, *Gli2*, *Gsk3b*, *Ptch2*, or *Smo* (*Figure 3D*; *Figure 3—figure supplement 1*). However, differentially accessible chromatin regions (DARs) were associated with *Gli1*, *Gli2*, and *Gsk3b* suggesting indirect regulation by Lhx2 (*Figure 3D*; *Figure 3—figure supplement 1*). In contrast, Lhx2 binding was associated with *Cdon*, *Gas1*, *Ptch1*, and *Gli3* (*Figure 3E F*; *Figure 3—figure supplement 1*), and several ChIP-seq peaks aligned with DARs, suggesting direct regulation by Lhx2 for these genes (*Figure 3E, F*; *Figure 3—figure supplement 1*; *Zibetti et al., 2019*). DARs, but not Lhx2 binding, were associated with other Shh pathway genes, but their low DESeq2 rankings suggest the predicted changes in chromatin accessibility are not functionally relevant (*Figure 3—figure supplement 2*). In sum, the expression of multiple Shh pathway genes is altered following *Lhx2*

inactivation, with several likely to be directly dependent on *Lhx2* and others more generally regulated by Shh signaling.

## Lhx2 does not regulate Shh bioavailability

Shh signaling in the retina is an example of *intra*-lineage signaling, where the Shh-producing cells (RGCs) are the direct descendants of the responding cells (RPCs). This configuration, in effect, places Shh upstream and downstream of the pathway (*Figure 4A*). Although RGCs are overproduced in the CKO and the expression of *Shh* and other genes involved in Shh production did not reach the cutoff for confident DEG designation (*Figure 3C*), it stands that mRNA expression levels are insufficient for predicting Shh bioavailability, especially given the importance of posttranslational mechanisms in the modification, secretion, and presentation of Shh to responding cells (*Briscoe and Thérond, 2013*). Furthermore, *Lhx2* loss of function could cause cryptic changes in extracellular factors that could negatively impact signaling. A more direct approach is to functionally test the bioavailability of endogenous Shh with minimal disruption to the extracellular environment. To do this, we adapted a biosynthetic system engineered to model gradient formation and signaling dynamics in Shh-responsive NIH3T3 cells (*Li et al., 2018*). In this reporter system, *Ptch1* is expressed under the control of a stably integrated doxycycline-regulated expression cassette in a *Ptch1* mutant background and signaling is reported by the expression of H2B-mCitrine under the control of 8 tandem GLI binding sites (*Figure 4B*). This configuration eliminates Ptch1-mediated negative feedback and expands the dynamic range of mCitrine reporter activity in these cells, referred to as *open-loop* cells. Important for our purposes, the *open-loop* cell line was designed to express *Ptch1* at a low level without doxy-cycline treatment, preventing Shh-independent activation of the reporter (*Li et al., 2018*). Therefore, this system has the potential to directly identify deficiencies in the bioavailability of endogenous Shh.

To assess the suitability of this reporter system, *open-loop* cells were grown to confluence without doxycycline followed by addition of recombinant human Shh (N-terminal fragment, C24II variant; referred to as Shh-N) and monitored for mCitrine fluorescence (*Figure 4—figure supplement 1A*). mCitrine + cells were rarely observed without Shh-N addition (*Figure 4—figure supplement 1*, top row). In contrast, duration and dose dependent increases in mCitrine expression and the number of mCitrine + *open* loop cells were observed after Shh-N addition (*Figure 4—figure supplement 1B, C*), and confirmed by the accumulation of mCitrine + cells by the sum of the product of their areas and mean fluorescence intensities at 72 hr (*Figure 4C*; *Supplementary file 4*). Addition of the Shh-blocking IgG antibody, 5E1, to the culture medium suppressed mCitrine expression (*Figure 4D*), demonstrating a continued requirement for Shh-N in the open-loop cells even at the lowest levels of Ptch1 expression (i.e. without doxycycline treatment). Taken together, these data indicate that mCitrine expression in the *open-loop* cells is dependent on Shh-N in a dose-dependent manner.

We next tested the ability of retinal tissue to stimulate signaling in *open-loop* cells. Prior work showed that mCitrine + *open* loop cells extended several cell diameters away from cellular sources of Shh as long as cells remained confluent (*Li et al., 2018*). We therefore developed a coculture para-digm in which whole retinal tissue was flat mounted onto the underside of a transwell insert, placing the retina in direct contact with a confluent monolayer of *open-loop* cells (*Figure 4E*). We initially tested the apical and basal surfaces of the E18.5 wild type retina because RGCs, the Shh-producing cells, are located at the basal surface and Shh signaling extends across the central to peripheral axis of the retina by this age (*Figure 4F*; *Sigulinsky et al., 2008*). As anticipated, *open-loop* cells responded preferentially when in contact with the basal retinal surface (*Figure 4G*). Taken together, our observa-tions show that this reporter system can be used to assess the bioavailability of endogenous Shh in the intact retina especially when the *open-loop* cells are in direct apposition to the basal retinal surface.

If reduced signaling was not due to impaired Shh activity or availability, then *open-loop* cells should respond similarly to CKO retinas compared to control. This response could be reflected in the number of mCitrine + cells, the expression levels of mCitrine, or in the kinetics of mCitrine accumulation. Tamoxifen was administered at E11.5, retinas were harvested at E15.5, placed onto confluent *open-loop* cells with the retinal basal surface in direct contact, and longitudinally imaged at 24 hr intervals by widefield epifluorescence (*Figure 4H*; single channel and bright field images for representative co-cultures are provided in *Figure 4—figure supplement 2*). mCitrine expression was not observed at the start of the culture period (*Figure 4H*, top row), but was readily apparent in *open-loop* cells cocultured with control (n=10) or CKO (n=13) retinas by 72 hr (*Figure 4H*, bottom row). mCitrine

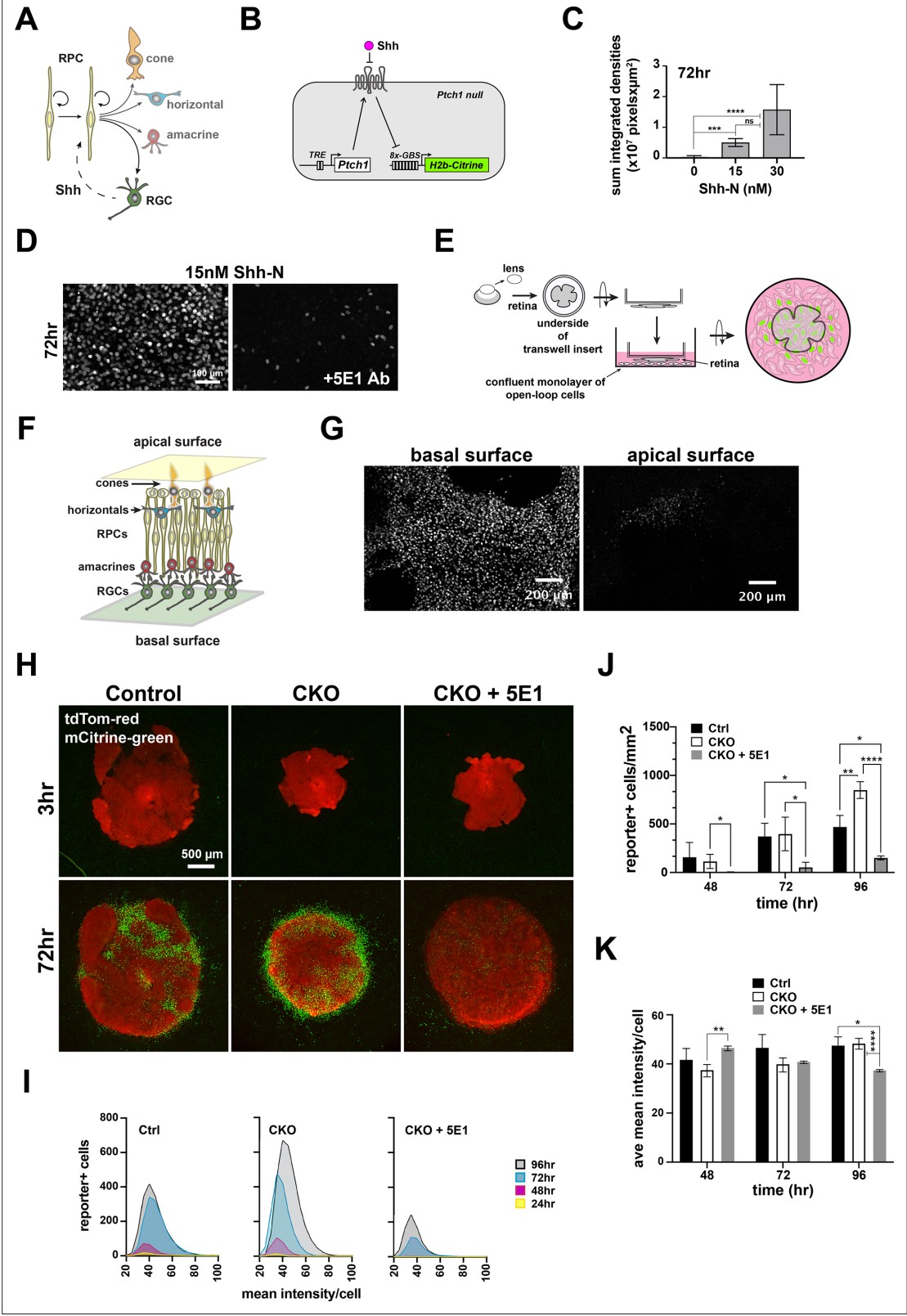

**Figure 4.** Evaluation of a live cell reporter system to test Shh bioavailability. (**A**) *Intra*-lineage architecture of Hh signaling at the start of retinal neurogenesis. RPCs initiate neurogenesis and begin generating RGCs, the Shh producing cells. RPCs are the responder cells, placing Shh upstream and downstream of RPCs. (**B**) Configuration of the *open-loop* circuit in the responder NIH3T3 *Ptch1*-null cell line (*open-loop* cells). Ptch1 is produced from a doxycycline regulated transgene. Shh binds Ptch1, activating intrinsic signaling as well as Gli-dependent expression of mCitrine fused to

*Figure 4 continued on next page*

*Figure 4 continued*

Histone H2b through 8-multimerized Gli1 binding sites. (**C**) Dose response at 72 hr as a function of the total accumulation of signal intensities and area coverage of mCitrine + *open* loop cells (mean+/-S.D.; n=4 per condition; ns, not significant; ***, $p_{adj}$ <0.001; ****, $p_{adj}$ <0.0001; ANOVA followed by Tukey's multiple comparisons; see *Supplementary file 4* for statistics). (**D**) Addition of the Shh ligand-blocking monoclonal antibody, 5E1, diminishes mCitrine expression. (**E**) Design of coculture experiment. Freshly dissected embryonic retina is flat mounted to the underside of a transwell insert and placed into direct contact with a confluent monolayer of *open-loop* cells. mCitrine expression accumulates in *open-loop* cells that receive Shh from the retina. (**F**) Schematic cross-section of embryonic retina shows that the apical surface is comprised mainly of RPCs and developing photoreceptors (cones until ~E15.5, and a mix of rods and cones thereafter) and the basal surface is comprised of RGCs, the source of retinal SHH. Astrocytes and endothelial also reside on the basal surface (not shown). (**G**) E18.5 wild type retinas were cultured in opposite orientations such that the *open-loop* cells contacted the basal or apical surfaces of the retina. mCitrine expression was robustly induced in responder cells in close proximity to the basal surface but not the apical surface of the retina. (**H**) Cocultures for control (left), CKO (middle), and CKO incubated with 5E1 antibody (right) at 3 and 72 hr. Retinal tissues are tdTomato positive (red) and mCitrine positive nuclei are green. (**I**) Lowess-smoothed histograms showing the accumulation and fluorescence intensity distributions of mCitrine + responder cells at each timepoint during the co-culture period. The histograms are for the cocultures shown in A. (**J**) Quantification of mCitrine + cells at 48, 72, and 96 hr. Comparisons were done within timepoints only and the significant differences are shown (mean+/-S.D.; n=4 (Ctrl, all timepoints); n=6 (CKO, all timepoints); n=2 (CKO+5E1, all timepoints); *, $p_{adj}$ <0.05; **, $p_{adj}$ <0.01; ****, $p_{adj}$ <0.0001; 2-way repeated measures ANOVA followed by Tukey's multiple comparisons test; see *Supplementary file 4* for statistics). (**K**) Quantification of the average mCitrine fluorescence intensities per cell at 48, 72, and 96 hr. Comparisons were done within timepoints only and the significant differences are shown (mean+/-S.D.; n=4 (Ctrl, all timepoints); n=6 (CKO, all timepoints); n=2 (CKO, all timepoints); *, $p_{adj}$ <0.05; **, $p_{adj}$ <0.01; ****, $p_{adj}$ <0.0001; 2-way repeated measures ANOVA followed by Tukey's multiple comparisons test; see *Supplementary file 4* for statistics).

The online version of this article includes the following figure supplement(s) for figure 4:

**Figure supplement 1.** Responder cells express mCitrine near the basal surface of the retina.

**Figure supplement 2.** Single channel images Single channel fluorescence and bright field images for the cocultures shown in *Figure 5B*.

expression was Shh dependent, revealed by the reduction in mCitrine + *open* loop cells when the 5E1 antibody was added to the culture medium (*Figure 4H*, n=2 explants). Using representative explants for each condition, the accumulation and fluorescence intensities of mCitrine + cells were plotted over time (*Figure 4I*). In general, the behavior of the *open-loop* cells exhibited similar temporal characteristics, but CKO explants supported the highest number of mCitrine + cells at 96 hr, and the accumulation of mCitrine + cells in the presence of 5E1 was notably reduced. Quantification of the total number of mCitrine + cells per explant area revealed that Ctrl and CKO explants promoted similar numbers of mCitrine + cells at 48 hr and 72 hr, but more mCitrine + cells were associated with the CKO retina at 96 hr (*Figure 4J*; *Supplementary file 4*). This higher number of mCitrine + cells could be due to a higher concentration or bioavailability of Shh in the CKO (see discussion). On a per sample (n) basis, the averages of the mean Tfluorescence intensities of the *open-loop* cells were generally similar between Ctrl and CKO explants, indicating similar levels of signaling were achieved (*Figure 4K*; *Supplementary file 4*). Subtle but significant differences in fluorescence intensities were observed with the 5E1 antibody (*Figure 4K*), but these differences could have been due to stochastic variation in Ptch1 expression in this small cohort of *open-loop* cells (*Li et al., 2018*). In sum, these data reveal that Shh bioavailability is not compromised in the CKO retina and point to a cell-autonomous role for *Lhx2* in RPCs to promote Shh signaling during early retinal neurogenesis.

## Modulating Smoothened activity ex vivo stimulates Shh signaling in Lhx2-deficient RPCs

We next asked if the reduced Shh signaling activity in the CKO retina was due to defective intracellular signal transduction. Smo is an obligate component of Shh signaling and functions at the interface of the extracellular and intracellular portions of the pathway (*Figure 1A*; *Briscoe and Thérond, 2013*). We reasoned that if the pathway was not functional at the level of Smo or downstream, attempts to stimulate signaling via Smo would fail. To address this, we utilized an ex vivo organotypic culture paradigm previously used to assess Shh signaling in postnatal day 0 (P0) RPCs (*Sigulinsky et al., 2008*). Here, explant cultures consisting of whole retina with the lens still attached were treated with purmorphamine, a small molecule agonist of Smo that bypasses Patched-mediated inhibition (*Stanton and Peng, 2010*; *Figure 5A*). To minimize the potential impact of the tissue phenotype on gene expression levels, the interval between tamoxifen treatment and the start of the culture was shortened to 3 days (tamoxifen at E11.5, tissue harvest at E14.5), which was still sufficient to observe reduced *Gli1* and *Smo* expression in the CKO (*Figure 1G*; *Supplementary file 3*).

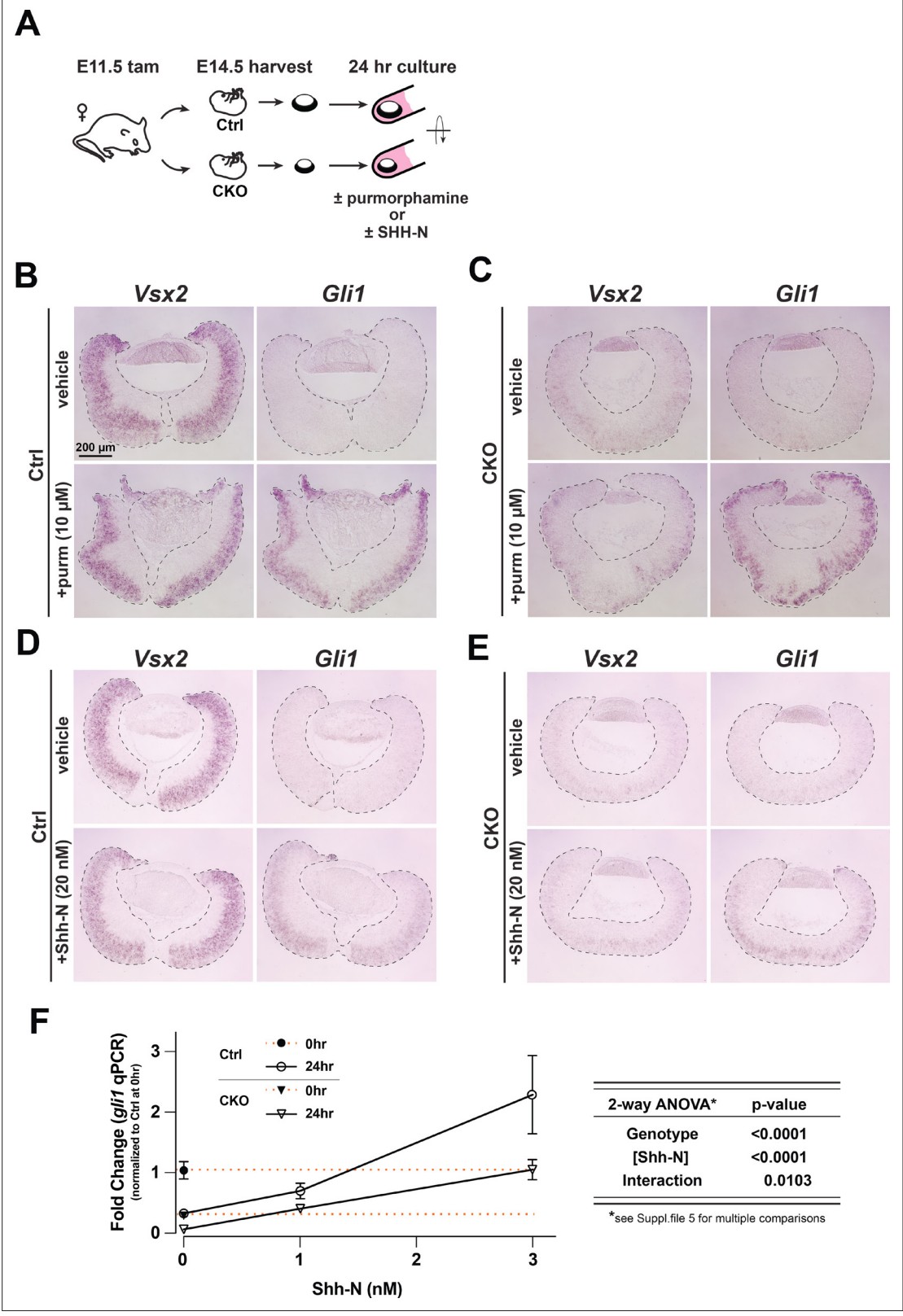

**Figure 5.** Purmorphamine and recombinant Shh-N stimulate signaling in *Lhx2* CKO retinal explants. (**A**) Experimental design of ex vivo retina-lens explant cultures. (**B–E**) in situ hybridizations of *Vsx2* and *Gli1* expression after 24 hr in culture to test *Smo* agonist purmorphamine or recombinant Shh-N. (**B**) *Gli1* expression declines in the absence of purmorphamine (vehicle) whereas *Vsx2* is maintained. *Gli1* expression is restored with purmorphamine. (**C**) *Vsx2* declines due to *Lhx2* inactivation. *Gli1* is expressed in response to purmorphamine. (**D**) *Gli1* expression declines in the absence of Shh-N

*Figure 5 continued on next page*

*Figure 5 continued*

(vehicle) whereas *Vsx2* is maintained. *Gli1* is restored with Shh-N. (**E**) *Vsx2* declines due to *Lhx2* inactivation. Similar to purmorphamine, *Gli1* expression is upregulated with Shh-N. (**F**) qPCR-based *Gli1* expression in control and CKO retinal explants at the start of the culture (t=0) and after 24 hr at different concentrations of Shh-N, as determined from a pilot dose response with wild type retinal explants (*Figure 5—figure supplement 1C*, *Supplementary file 5*). Expression values are relative to the mean control value at t=0 (closed circle). Orange lines extend from t=0 values for the control (upper line) and the CKO (closed triangle, lower line). Note that the value for the 24 hr control in 0 nM Shh-N overlaps with the CKO at t=0. To the right of the graph is the summary table for two-way ANOVA showing that the main effects (genotype and Shh-N concentration) are significant and interact. mean+/-S.E.M.; n=4 all conditions except CKO, 1nM Shh-N (n=3); Two-way ANOVA followed by Tukey's multiple comparisons test; See *Supplementary file 5* for statistics including $p_{adj}$ values for multiple comparisons.

The online version of this article includes the following figure supplement(s) for figure 5:

**Figure supplement 1.** Markers of recombination, apoptosis, and RGCs after 24 hr, and dose response to estimate physiological range for recombinant Shh-N(C24II) protein.

After 24 hr in culture, *Gli1* expression was markedly reduced in the vehicle-treated explants compared to explants treated with 10 µM purmorphamine (*Figure 5B*, right images). Since the RPC gene *Vsx2* was still abundantly expressed (*Figure 5B*, left images), the drop in *Gli1* expression in the untreated control explants was not due to RPC loss but instead to a specific reduction in Shh signaling. RGCs exhibited enhanced apoptosis (*Figure 5—figure supplement 1A, B*), potentially reducing the availability of endogenous Shh (*Wang et al., 2002*). We therefore attribute the robust expression of *Gli1* in the treated control explants to purmorphamine. Interestingly, *Gli1* expression was activated in CKO explants treated with purmorphamine (*Figure 5C*). Since the initial level of *Gli1* in the CKO explants was already reduced due to *Lhx2* inactivation, this result indicates that pathway activation at the level of Smo can occur in the absence of *Lhx2*. We also treated retinal explants with 20 nM Shh-N and obtained similar results (*Figure 5D and E*). These observations support the idea that the Shh pathway downstream of Patched and Smo is functional in the absence of *Lhx2*.

To determine if CKO RPCs respond differently than control RPCs to Shh-N treatment, we assessed *Gli1* expression by qPCR in a pilot dose response experiment with wild type E14.5 retinal explants cultured for 24 hr (*Figure 5—figure supplement 1C*). As expected, retinal explants cultured without Shh-N exhibited a strong reduction in *Gli1* expression compared to its level at the start of the experiment (t=0), whereas 3 nM Shh-N was sufficient to maintain signaling, and 10 nM significantly enhanced signaling (*Figure 5—figure supplement 1C*; *Supplementary file 5*). Based on these observations, we tested the responsiveness of CKO and control explants at 0, 1, and 3 nM Shh-N (*Figure 5F*). With all expression data normalized to the level of *Gli1* expression in control explants at t=0 (solid circle, upper dashed orange line), we made several observations (statistics are listed for all comparisons in *Supplementary file 5*). First, when compared to CKO explants at t=0 (closed triangle, lower dashed red line), *Gli1* expression decreased in CKO explants cultured without Shh-N (0 nM Shh-N; open triangle,), revealing that endogenous Shh promotes a low level of signaling in the CKO retina. Second, *Gli1* expression dropped in control explants cultured without Shh-N (open circle, 0 nM Shh-N) to the same level as the CKO retina at t=0 (closed triangle, lower dashed red line). This is consistent with the low level of *Gli1* expression in the vehicle treated control explants shown in *Figure 5B and D*. Third, CKO and control explants exhibited a similar response profile to Shh-N from 0 to 1 nM, but CKO explants exhibited a reduced response from 1 to 3 nM. Two-way ANOVA showed significant differences in both genotype and Shh-N concentrations as well as an interaction between both (*Figure 5F*; *Supplementary file 5*). These data indicate that endogenous Shh is promoting a low level of signaling in *Lhx2*-deficient RPCs, that *Lhx2*-deficient RPCs can respond to recombinant Shh-N at more physiologically relevant concentrations, but their response is still attenuated compared to *Lhx2*-expressing RPCs.

## Ptch1 inactivation stimulates Shh signaling in the Lhx2-deficient retina but fails to restore retinal development

We next tested if Shh signaling could be stimulated in *Lhx2* CKO RPCs by the simultaneous removal of *Ptch1* function. Conditional *Ptch1* inactivation in the embryonic limb caused increased and ectopic Shh signaling as revealed by elevated readout expression and Shh gain of function phenotypes (*Butterfield et al., 2009*). Like purmorphamine, *Ptch1* inactivation exposes the activity of endogenous Smo in a Shh-independent manner and provides a cell autonomous, in vivo test of signaling competence at

the level of *Smo* in *Lhx2*-deficient RPCs. It also allowed us to directly test whether Ptch1 is restricting signaling in the *Lhx2* CKO retina and, if so, whether its inactivation improves retinal development.

*Ptch1* CKO, *Lhx2* CKO, and *Ptch1, Lhx2* double CKO (dCKO) retinas were generated with the *Hes1^{CreER}* driver (**Figure 6—figure supplement 1A**). Retinas were harvested at E15.5 from embryos treated with tamoxifen at E10.75 and E11.5, and recombination of the *Ptch1^{flox}* allele was assessed by RT-PCR with primers that amplify both the intact *floxed* and *deleted* transcripts (**Butterfield et al., 2009**; **Figure 6—figure supplement 1B, C**). Non-recombined transcript was detected but its relative abundance was low compared to the deleted transcript. Therefore, all samples were used to measure the relative expression levels of *Gli1, Ptch1, Ptch2,* and *Hhip* by qPCR (**Figure 6A**). By one-way ANOVA, the main effect of genotype on all four genes was highly significant (p<0.0001), and for each gene, at least four of the six pairwise genotype comparisons showed significant differences in expression (**Supplementary file 3**). Of those, *Hhip* and *Ptch1* expression were increased in the *Ptch1* CKO compared to control, indicating that *Ptch1* inactivation on its own stimulated Shh signaling. *Gli1* and *Ptch2* expression were decreased in the *Lhx2* CKO, consistent with their high DEG rank in the RNA sequencing data. Importantly, the expression levels of all four genes were significantly higher in the dCKO compared to the *Lhx2* CKO, with *Ptch1, Hhip,* and *Ptch2* surpassing control levels (**Figure 6A**). These data provide in vivo evidence that *Lhx2*-deficient RPCs retain the competence to signal at the level of *Smo*, and that *Ptch1* is inhibiting signaling in the absence of *Lhx2*.

Despite the evidence for increased Shh signaling, the histogenesis defects due to *Lhx2* inactivation did not improve with *Ptch1* inactivation (**Figure 6B**). DAPI staining revealed persistent disrupted retinal organization. PCNA staining and EdU incorporation revealed a persistent deficit of RPCs, and the disorganized distribution of Pou4f2+RGC and Otx2 + photoreceptor precursors were consistent with disrupted tissue cytoarchitecture. The failure of *Ptch1* inactivation to alleviate, even partially, the *Lhx2* CKO phenotype suggested that Lhx2 also acts downstream of Shh signaling. To assess this further, we examined the expression of Cyclin D1 and Hes1, which are both highly ranked DEGs in the *Lhx2* CKO, are regulated by Shh signaling (**Sakagami et al., 2009**; **Wang et al., 2005**; **Hashimoto et al., 2006**; **Kenney and Rowitch, 2000**; **Wall et al., 2009**) and are required for retinal neurogenesis (**Bosze et al., 2020**; **Das et al., 2009**; **Das et al., 2012**; **Takatsuka et al., 2004**). As with PCNA and EdU, reductions in Hes1 + and Cyclin D1 + cells were similar in dCKO and *Lhx2* CKO retinas when compared to control (**Figure 6C and E**). However, Cyclin D1 + cells appeared brighter in the mutant retinas and was confirmed with fluorescence intensity measurements (**Figure 6D**). Hes1 + cells also appeared brighter, but more so in the dCKO retina (**Figure 6F**). These changes in cellular fluorescence intensities suggest increased expression, and at least for Hes1, appears to be dependent on *Ptch1* inactivation. The lack of phenotypic rescue suggests that elevated Shh signaling due to *Ptch1* inactivation extended to Hes1 but was insufficient to improve retinal development.

## Gas1 and Cdon mediate Lhx2-dependent activation of Shh signaling

While our data show that Lhx2 influences the expression of multiple Shh pathway genes, measurable increases in pathway activity were still achieved in the *Lhx2* CKO with Shh-N or purmorphamine treatment in vitro, and *Ptch1* inactivation in vivo. These findings indicate that *Lhx2* inactivation did not cause an insurmountable block in the intracellular portion of the pathway and raises the possibility that an additional level of the pathway is dependent on *Lhx2*. Since reduced Shh availability was effectively ruled out, this leaves receptivity to Shh, possibly at the level of co-receptor function. Interestingly, the Shh co-receptors *Boc, Cdon, Gas1,* and *Lrp2* are expressed in the embryonic retina, but *Boc* and *Lrp2* are unlikely candidates because their ranks for differential expression in the RNA-seq dataset were far below the cutoff (10530 and 11268, respectively) and their genetic inactivation largely spares early retinal neurogenesis (**Cases et al., 2015**; **Fabre et al., 2010**). On the other hand, both *Gas1* and *Cdon* qualified as DEGs with DESeq2 ranks of 117 and 1248, respectively, and the ChIP- and ATAC-seq data support direct gene regulation by *Lhx2* (**Figure 3D and E**). in situ hybridization and qPCR confirmed their downregulation in the E15.5 *Lhx2* CKO retina following tamoxifen treatment at E11.5 (**Figure 7—figure supplement 1A**, **Supplementary file 3**), but their expression at E15.5 in the control retina was limited to the retinal periphery (**Figure 7—figure supplement 1A**, insets). This is not unexpected since Gas1 and Cdon are negatively regulated by Shh signaling in other tissues (**Allen et al., 2007**; **Tenzen et al., 2006**), but their restricted expression at E15.5 made it difficult to determine the extent of their dependence on Lhx2 in the retina. We therefore examined

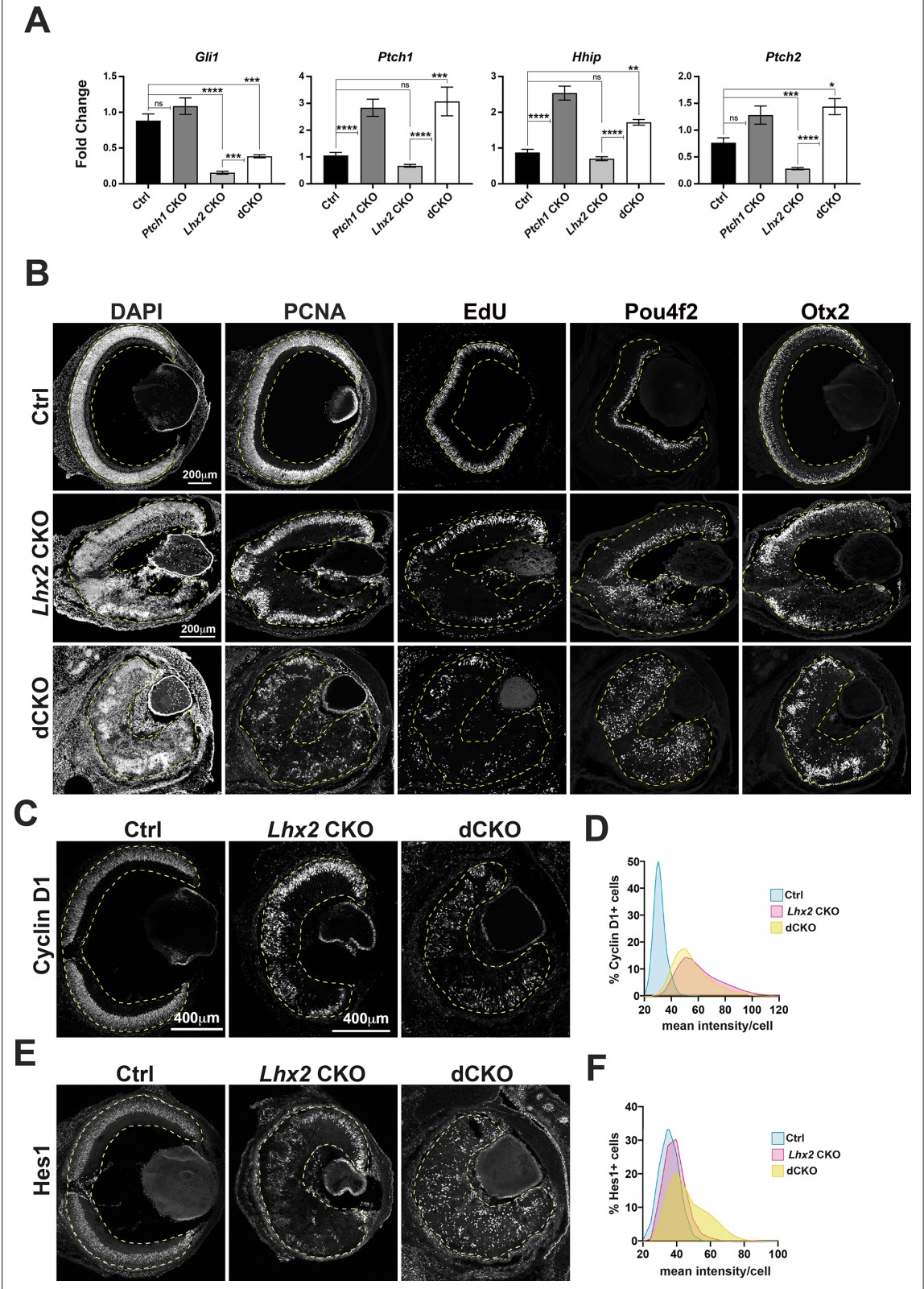

**Figure 6.** Hh signaling is enhanced in *Lhx2*-deficient RPCs by *Ptch1* inactivation in vivo. (**A**) Relative expression of *Gli1*, *Ptch1*, *Hhip*, and *Ptch2* at E15.5 following tamoxifen treatment at E11.5 in retinas with the following genotypes: control, *Ptch1* CKO, *Lhx2* CKO and *Lhx2; Ptch1* double CKO (dCKO). See *Figure 6—figure supplement 1A* for breeding scheme and genotypes assigned to control. For each gene, fold change values are relative to a specific control sample set as a reference. Shown are the comparisons for the three mutant genotypes compared to control and for the dCKO compared to the

*Figure 6 continued on next page*

Figure 6 continued

*Lhx2* CKO (means+/-S.E.M.; n=6 (Ctrl, all genes); n=9 (*Ptch1* CKO, all genes); n=7 (*Lhx2* CKO, all genes); n=4 (dCKO, all genes); ns, not significant; * $p_{adj}$ <0.05; ** $p_{adj}$ <0.01; *** $p_{adj}$ <0.001; **** $p_{adj}$ <0.0001; ANOVA followed by Tukey's multiple comparisons test; see *Supplementary file 6* for statistics and complete multiple comparisons list). (**B**) DAPI staining and expression patterns for PCNA and EdU incorporation to identify RPCs and Pou4f2 and Otx2 to identify nascent RGCs and photoreceptors at E18.5 following tamoxifen treatment at E11.5 in retinas from control (top row), *Lhx2* CKO (middle row) and dCKO (bottom row). (**C**) Cyclin D1 expression at E18.5 following tamoxifen treatment at E11.5 in control, *Lhx2* CKO and dCKO retinas. (**D**) Lowess-smoothed histograms showing the distribution of Cyclin D1 + cells as a function of the mean fluorescence intensity per cell. Each histogram is normalized to the number of Cyclin D1 + cells within the respective genotype. (**E**) Hes1 expression at E18.5 following tamoxifen treatment at E11.5 in control, *Lhx2* CKO and dCKO retinas. (**F**) Lowess-smoothed histograms showing the distribution of Hes1 + cells as a function of the mean fluorescence intensity per cell. Each histogram is normalized to the number of Hes1 + cells within the respective genotype.

The online version of this article includes the following source data and figure supplement(s) for figure 6:

**Figure supplement 1.** Genetics of *Ptch1* and *Lhx2* inactivation and validation of *Ptch1* recombination.

**Figure supplement 1—source data 1.** RT-PCR for non-deleted (*flox*) and deleted (Δ) *Ptch1* mRNAs.

their expression patterns from E11.5 – E13.5, the interval encompassing pathway activation. By both immunohistology (*Figure 7A*) and in situ hybridization (*Figure 7—figure supplement 1B*), *Gas1* was more restricted to the peripheral retina, but *Cdon* was broadly expressed at E11.5, resolving to the peripheral retina by E13.5. *Cdon* expression decreased in a complementary manner to *Gli1* upregulation as revealed by *Gli1* mRNA (*Figure 7—figure supplement 1B*) and by b-Galactosidase (b-Gal) reporter expression from the *Gli1^lacz^* allele (*Figure 7A*). However, *Cdon* downregulation appeared to be ahead of the central to peripheral wave of *Gli1* expression and had a closer complementarity with *Atoh7* (*Figure 7—figure supplement 1B*), a neurogenic bHLH gene transiently expressed in RPCs that functions as an RGC competence factor (*Brown et al., 2001*; *Brzezinski et al., 2012*; *Prasov and Glaser, 2012*). Whether this reflects a novel mode of *Cdon* regulation is unclear, but the complementarity with *Cdon* and *Gli1* expression is consistent with downregulation upon pathway activation.

We next examined *Gas1* and *Cdon* expression in the CKO by in situ hybridization at E12.5 (*Figure 7—figure supplement 1C*) and immunohistology at E13.5 (*Figure 7B*) following tamoxifen treatment at E10.5 and E11.5, respectively. At E12.5, the *Lhx2* target *Vsx2* was downregulated as were *Gas1* and *Cdon*, consistent with *Lhx2* promoting their expression, but it was too early to assess effects on *Gli1* since it was not yet detected in the control retina (*Figure 7—figure supplement 1C*). At E13.5, Gas1 and Cdon proteins were downregulated, and b-Gal expression from the *Gli1^lacz^* allele was not detected in the CKO (*Figure 7B*). These observations support the idea that Lhx2 promotes the expression of *Cdon* and *Gas1* to confer signaling competence to RPCs. Interestingly, *Cdon* mRNA and protein expression persisted to some extent in the dorsal CKO retina at both ages (*Figure 7B*; *Figure 7—figure supplement 1C*). Whether this reflects mechanistic differences in Shh signaling or differences in how Lhx2 regulates Shh signaling across the retina is unclear.

To determine if *Gas1* and *Cdon* were sufficient to restore signaling, we overexpressed *Gas1* and *Cdon* and assessed their effects on b-Gal reporter expression from the *Gli1* locus (*Figure 8*). *Lhx2* CKO; *Gli1^LacZ^* explants were electroporated with *pCIG* or *pCIG-Gas1* and *pCIG-Cdon* at the start of the culture followed by addition of 3 nM Shh-N at 24 hr and cultured for an additional 24 hr (*Figure 8A and B*). Since electroporation is not cell type specific, 1 μM EdU was also added to label proliferating RPCs. Cdon and Gas1 were not detected in CKO explants transfected with *pCIG* alone (*Figure 8C*, upper panels; but were readily detected in explants co-transfected with *pCIG-Gas1* and *pCIG-Cdon* (*Figure 8C*, lower panels; *Figure 8—figure supplement 1* contains single channel images for all panels)). Triple labeling for GFP (green), EdU (blue), and b-Gal (red) revealed a significant increase in b-Gal + reporter cells in explants co-transfected with *pCIG-Gas1* and *pCIG-Cdon* compared to explants transfected with *pCIG* only (*Figure 8D and E*; *Supplementary file 6*). The increase in b-Gal + cells is not likely to be due to escaper cells (i.e. Lhx2 + cells) since pCIG transfected cells were rarely positive for Lhx2 and b-Gal in CKO explants compared to control explants (*Figure 8F*; *Figure 8—figure supplement 1E*). These data support the hypothesis that *Lhx2* confers signaling competence in RPCs through promoting *Cdon* and/or *Gas1* expression.

## Lhx2 promotes Shh signaling after Gas1 and Cdon downregulation

Although re-expressing *Cdon* and *Gas1* increased expression from the *Gli1* locus in the absence of *Lhx2*, it remains that reaching or maintaining the appropriate level of signaling could still depend

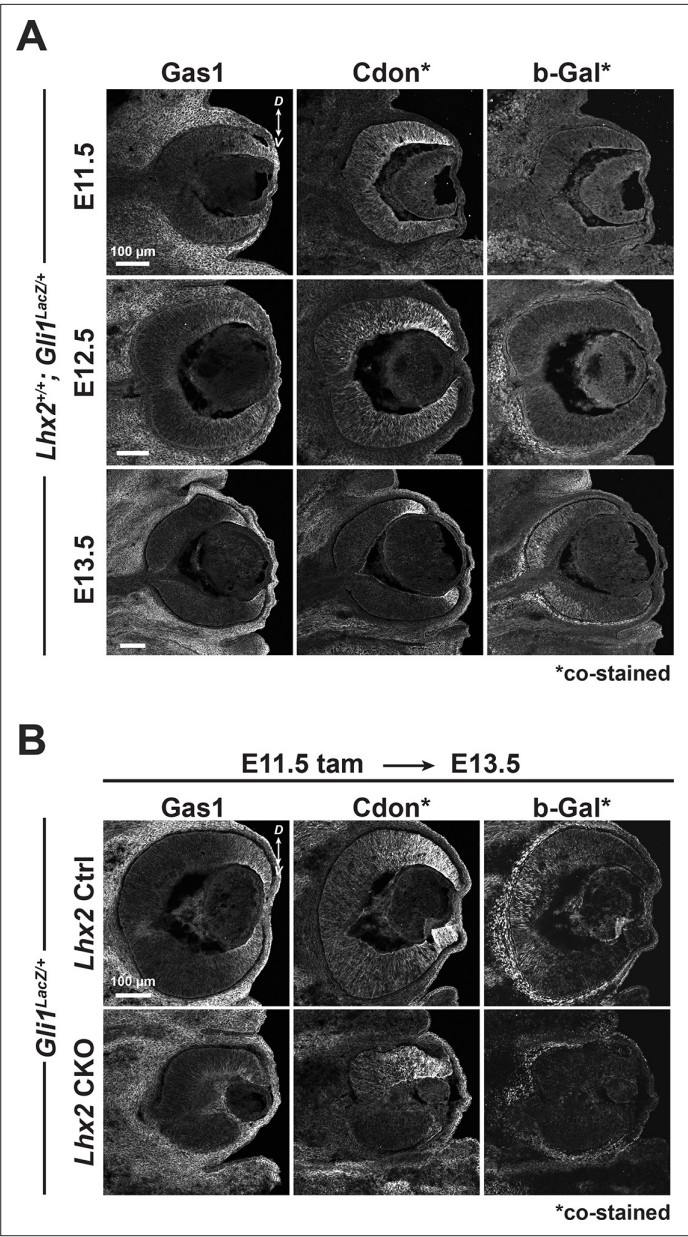

**Figure 7.** Cdon and Gas1 expression are dependent on Lhx2 prior to their downregulation at the start of Shh signaling. (**A**) Temporal expression patterns of Gas1, Cdon, and β-Gal at E11.5, E12.5, and E13.5 in *Gli1^{lacz/+}* mice. Cdon and β-Gal were detected on the same tissue sections, Gas1 on adjacent sections (also in B). (**B**) Expression of Gas1, Cdon, and β-Gal in Ctrl and *Lhx2* CKO; *Gli1^{lacz/+}* eyes at E13.5 following tamoxifen treatment at E11.5.

The online version of this article includes the following figure supplement(s) for figure 7:

**Figure supplement 1.** in situ hybridizations.

on regulation of other Shh pathway genes by Lhx2. Since *Cdon* and *Gas1* are downregulated by E13.5, inactivating *Lhx2* after E13.5 provided an opportunity to test this. Tamoxifen was administered at E14.5 and retinas collected at E17.5 to assess changes in relative gene expression of *Gli1, Gli2, Ptch2,* and *Smo* by qPCR (*Figure 9A*; *Supplementary file 3*). *Lhx2* downregulation was highly efficient and accompanied by the predicted drop in *Vsx2*. Interestingly, *Gli1, Gli2, Ptch2,* and *Smo* were also reduced in the CKO retina. These changes were not due to developmental disruptions caused by *Lhx2* inactivation because early retinal neurogenesis is largely spared with tamoxifen treatment by E13.5 (*Gordon et al., 2013*). Furthermore, the reductions in *Gli2, Ptch2,* and *Smo* are not strictly due to reduced *Gli1* activity because their expression levels were not significantly altered in the *Gli1*

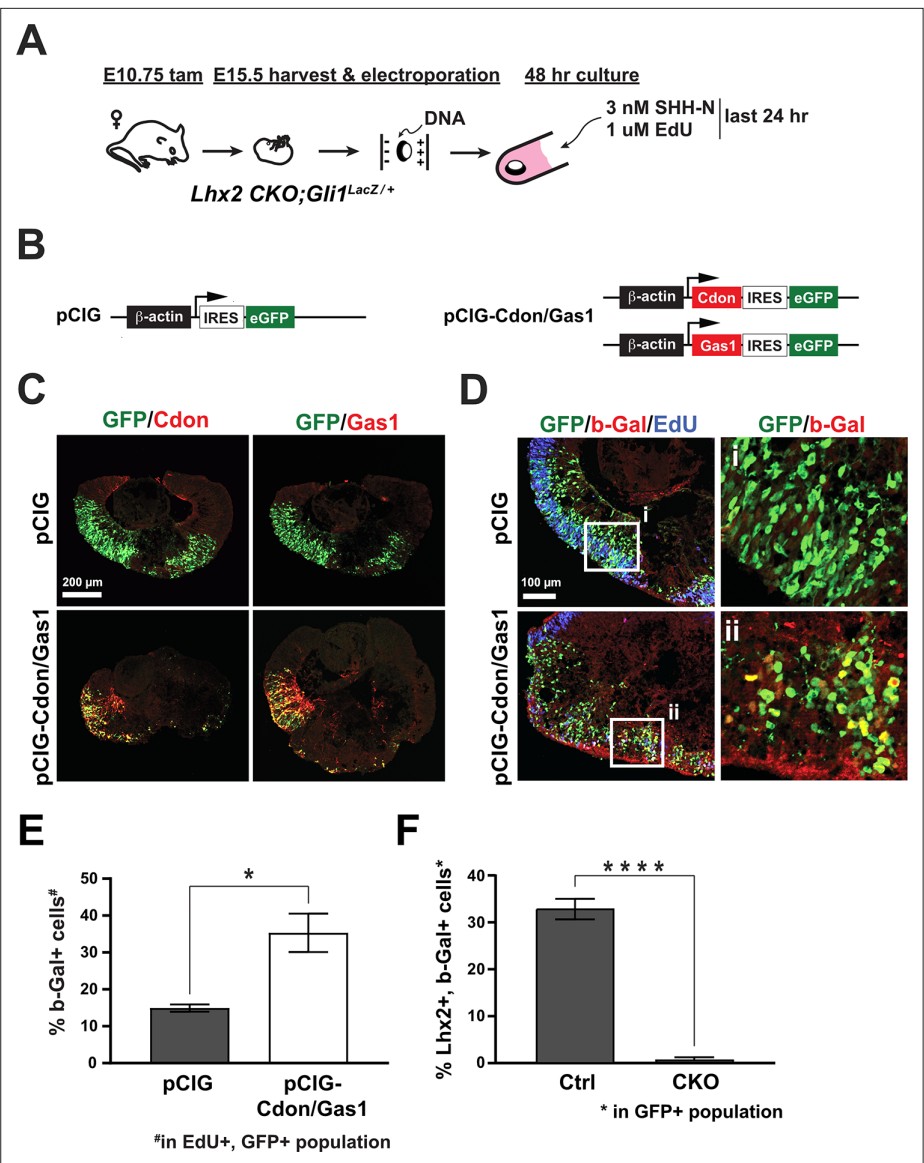

**Figure 8.** Cdon and Gas1 overexpression is sufficient to stimulate Shh signaling in the absence of Lhx2. (**A**) Experimental design for ex vivo electroporation and explant culture. *Lhx2* CKO; *Gli1^LacZ/+^* explants were electroporated at the beginning of the culture. 3 nM Shh-N and 1 µM EdU were added after 24 hr and cultured for an additional 24 hr. (**B**) DNA constructs used for electroporation. pCIG served as the control and pCIG-Cdon and pCIG-Gas1 were co-electroporated. (**C**) Upper panels: explants were electroporated with pCIG and co-stained for GFP and Cdon (left panel) or Gas1 (right panel). Lower panels: explants were co-electroporated with pCIG-Cdon and pCIG-Gas1 and co-stained for GFP and Cdon (left panel) or Gas1 (right panel). (**D**) Electroporated explants were co-stained for GFP, β-Gal, and EdU. Insets (**i and ii**) show GFP and β-Gal staining only. (**E**) Quantification of the percentage β-Gal+ cells in the EdU+, GFP+ cell populations from GFP (control) and Cdon/Gas1 electroporated *Lhx2* CKO; *Gli1^LacZ/+^* explants (mean+/-S.E.M.; n=3, both conditions; *, p<0.05; n=3, both conditions; unpaired t-test). (**F**) Quantification of the percentage of Lhx2+, β-Gal+ cells in the GFP+ cell populations from control (*Lhx2^Δ/+^*; *Gli1^LacZ/+^*) and *Lhx2* CKO; *Gli1^LacZ/+^* explants electroporated with pCIG (mean+/-S.D.; n=7 (Ctrl); n=4 (CKO); *, p<0.0001; unpaired t-test) See **Supplementary file 6** for statistics. Single channel images for C and D and representative images for Ctrl (*Lhx2^Δ/+^*; *Gli1^LacZ/+^*) and *Lhx2* CKO; *Gli1^LacZ/+^* explants used for quantification in F are presented in *Figure 8—figure supplement 1*.

The online version of this article includes the following figure supplement(s) for figure 8:

**Figure supplement 1.** Single channel images for *Figure 8C and D*.

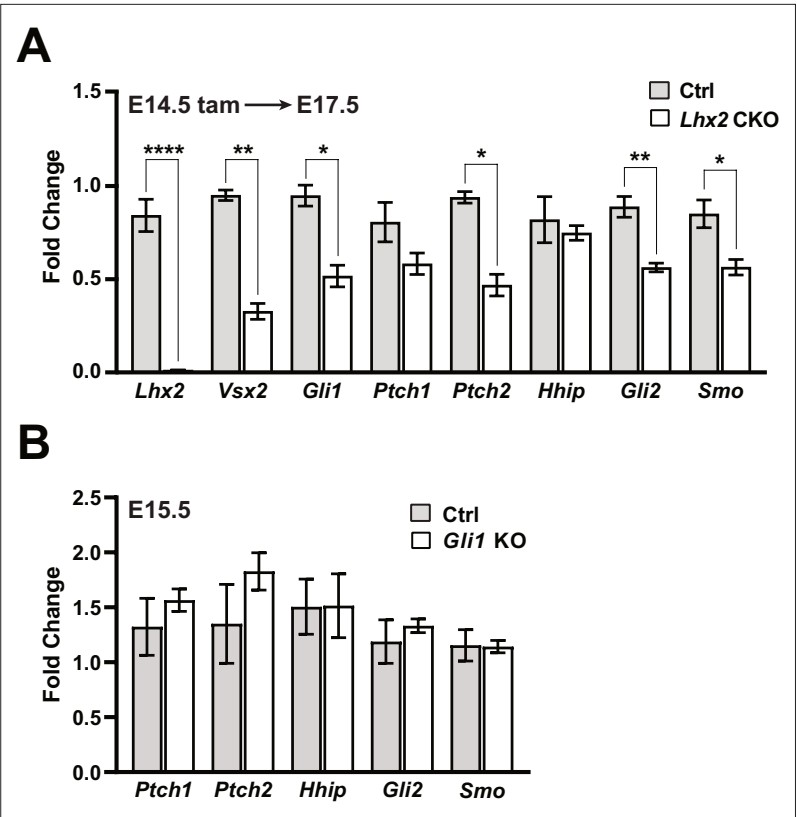

**Figure 9.** Lhx2 is required for sustained Shh signaling. (**A**) qPCR-based expression for *Lhx2, Vsx2, Gli1, Ptch1, Ptch2, Hhip, Gli2,* and *Smo* from E17.5 control and *Lhx2* CKO retinas following tamoxifen treatment at E14.5. For each gene, fold change values are relative to a specific control sample set as a reference. Only significant comparisons are noted (mean+/-S.E.M.; n=3 (Ctrl, all genes); n=4 (CKO, all genes); * $p_{adj}$ <0.05; ** $p_{adj}$ <0.01; **** $p_{adj}$ <0.0001; unpaired t-tests with correction for multiple comparisons; see ***Supplementary file 3*** for statistics) (**B**) qPCR-based expression for *Ptch1, Ptch2, Hhip, Gli2,* and *Smo* from E15.5 control (*Gli1$^{LacZ/+}$*) and *Gli1* KO retinas. For each gene, fold change values are relative to a specific control sample set as a reference. None of the comparisons were significant (mean+/-S.E.M; n=4 (both conditions, all genes); unpaired t-tests with correction for multiple comparisons; see ***Supplementary file 3*** for statistics).

KO retina at E15.5 (***Figure 9B***; ***Supplementary file 3***). From this, we conclude that *Lhx2* promotes Shh signaling at more than one point in the pathway with a measurable influence on signaling that is independent of *Cdon* and *Gas1*.

## Discussion

### Lhx2 is a multilevel modulator of the Shh pathway

*Lhx2* and Shh signaling are essential regulators of vertebrate retinal development that function at multiple stages. Here, we present evidence supporting a model in which Lhx2 promotes the expression of multiple genes in the Shh pathway that allows for the timely activation and sustained levels of signaling during early retinal neurogenesis by conferring signaling competence to RPCs (***Figure 10A***). Directly supporting this role for Lhx2, we show that endogenous Shh was functional and available in the *Lhx2*-deficient retina. Rather, Lhx2 supports ligand reception by what is likely direct regulation of expression of the co-receptors *Cdon* and *Gas1* and efficient signaling by promoting *Smo* and *Gli2* expression, possibly through indirect mechanisms (***Figure 10B***). As revealed by Shh-N treatment or *Ptch1* inactivation in the *Lhx2*-deficient retina, *Gli1* expression remains Shh-dependent, but in the specific context of the *Lhx2*-deficient retina, the reduced expression of *Gli1* could combine with reductions in *Gli2* and *Smo* to negatively impact sustained Shh signaling. Since Lhx2 regulates a broad repertoire of RPC genes at the epigenetic and transcriptional level (***Gueta et al., 2016***; ***Yun et al.,***

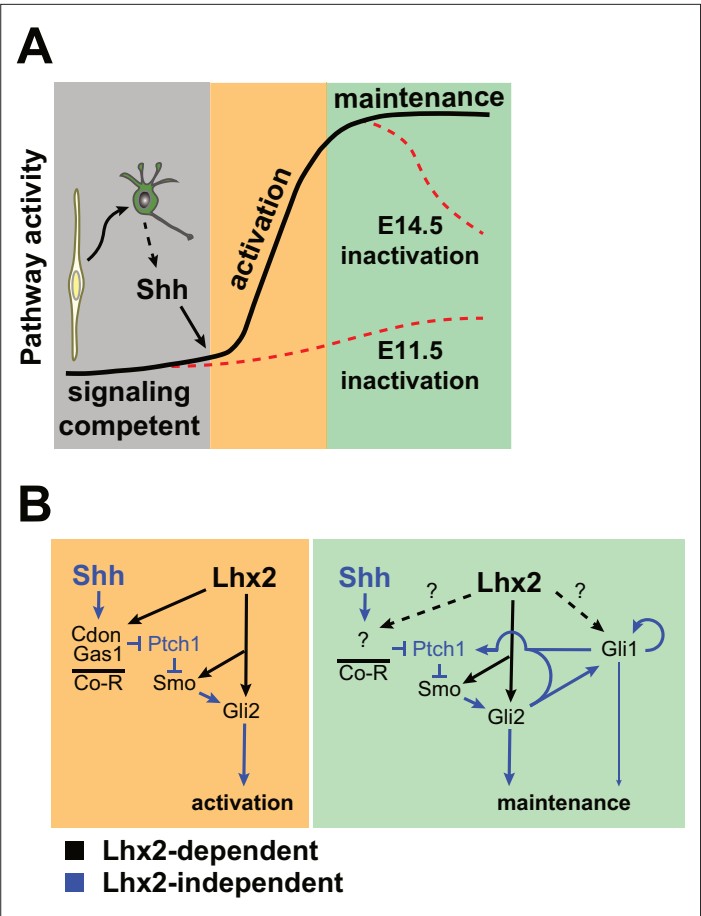

**Figure 10.** Models of interaction between *Lhx2* and Shh signaling during embryonic retinal neurogenesis. (**A**) At the cellular level, *Lhx2* promotes signaling competence in RPCs and once neurogenesis begins, RGCs produce Shh leading to pathway activation. This is revealed by *Lhx2* inactivation at E11.5. *Lhx2* also promotes the correct level of signaling in RPCs as evidenced by the drop in pathway readout gene expression when *Lhx2* is inactivated at E14.5. (**B**) Mechanistically, Lhx2 promotes signaling competence and efficient activation by promoting the expression of the coreceptors (Co-R) for ligand reception, *Smo* for signal transduction, and *Gli2* for target gene activation. During the maintenance phase, *Lhx2* promotes signaling again by promoting *Smo* and *Gli2*. Lhx2 may also regulate other coreceptors, *Gli1*, or other factors to promote efficient signaling.

*2009*; *Zibetti et al., 2019*), Lhx2 could also link Shh signaling to its downstream transcriptional targets as suggested by the failure of *Ptch1* inactivation to alleviate, even partially, the phenotypic consequences of *Lhx2* inactivation on early retinal neurogenesis. Based on the sum of our observations, we propose that Lhx2, in addition to promoting Shh signaling in RPCs, integrates the pathway into the program of early retinal neurogenesis.

A definitive link between Lhx2 and Shh signaling was established in the developing limb, where *Lhx2* (redundantly with *Lhx9* in mice) is required for Shh signaling during limb patterning and outgrowth (*Tzchori et al., 2009*; *Watson et al., 2018*). However, in the mouse limb, *Lhx2* and *Lhx9* (*Lhx2* only in chick) regulate the expression of *Shh* in the ZPA rather than the competence of the limb bud mesenchyme to respond (*Tzchori et al., 2009*; *Watson et al., 2018*). This is different from what is reported here, and further emphasizes how a multifunctional regulator such as *Lhx2* can influence developmental mechanisms (i.e. signaling) in a contextually specific manner.

## Lhx2 promotes Shh pathway activation by promoting Cdon and Gas1 expression

Our data indicate that *Lhx2* interacts with the Shh pathway in a complex manner, but the links to *Cdon* and *Gas1* are likely to transmit the largest impact on pathway activation. These linkages are revealed

by the rapid loss of *Cdon* and *Gas1* expression after *Lhx2* inactivation and by ChIP-seq data showing Lhx2 binding at the *Cdon* and *Gas1* loci. Upstream of *Cdon* and *Gas1* in the Shh pathway, we show that Shh availability was intact, and immediately downstream, purmorphamine treatment and *Ptch1* inactivation stimulated signaling in the *Lhx2*-deficient retina, as did *Cdon* and *Gas1* overexpression. These observations all point to a role for *Lhx2* in ligand reception, a function fulfilled by Shh co-receptors such as *Cdon* and *Gas1*.

This seemingly straightforward requirement for *Cdon* and *Gas1* in *Lhx2*-mediated Shh signaling contrasts with other identified roles for the co-receptors in early eye and retinal development. Prior to the onset of retinal neurogenesis, *Cdon* is expressed in the optic vesicle and functions in a manner consistent with both promoting and inhibiting Shh signaling (*Gallardo and Bovolenta, 2018*). A positive role for *Cdon* is revealed in *Cdon* KO mice, which exhibit a range of phenotypes consistent with absent or impaired Shh signaling, including holoprosencephaly (HPE) and Septo Optic Dysplasia (SOD) (*Bae et al., 2011*; *Cavodeassi et al., 2019*; *Kahn et al., 2017*; *Zhao et al., 2012*). While these early and severe phenotypes would normally preclude an assessment of its later role in RPCs, retinal development occurs in *Cdon* mutant mice that evade the HPE and SOD phenotypes (*Zhang et al., 2009*). In this context, the *Cdon* mutant phenotypes align well with retinal-specific *Shh* loss of function mutants, and similar to what we show here for *Lhx2* inactivation, *Cdon*-deficient RPCs fail to express *Gli1* even though *Shh* is still expressed (*Kahn et al., 2017*). Inhibitory effects of *Cdon* on Shh signaling was revealed in chick and zebrafish optic vesicles, where it limits the range of Shh signaling through ligand sequestration in signaling-*incompetent* optic neuroepithelial cells (*Cardozo et al., 2014*). An inhibitory role for another co-receptor, *Lrp2*, occurs in the nascent ciliary epithelium, where it is proposed to mediate endocytic clearance of Shh and prevent binding to Ptch1 protein (*Christ et al., 2015*). The nascent ciliary epithelium arises from the peripheral edge of the retinal neuroepithelium, the same location where *Gas1* and *Cdon* were downregulated at E15.5 following *Lhx2* inactivation at E11.5 (*Figure 7—figure supplement 1A*). Since Shh signaling is ectopically activated in this domain in *Lrp2* mutant mice (*Christ et al., 2015*), a primary role for *Lrp2* in the developing ciliary epithelium could be to prevent *Cdon* and *Gas1* from activating the pathway. Based on the sum of these findings, we propose that *Lhx2* sets up the signaling competence of RPCs by promoting *Cdon* expression throughout the retina and *Gas1* expression in the peripheral retina. This is countered by *Lrp2* in the far retinal periphery, where suppression of Shh signaling is required for ciliary epithelium development.

How *Gas1* relates to Shh signaling in early eye development is less clear. During optic vesicle patterning, *Gas1* is expressed in the nascent retinal pigment epithelium (RPE) and in *Gas1* KO mice, the predominant phenotype is an RPE to neural retina transformation in the ventral optic cup (*Lee et al., 2001*). This does not phenocopy the effects of directly disrupting Shh signaling, where inhibition reduces the proximal and ventral domains of the optic vesicle, and overactivation expands them (*Amato et al., 2004*; *Cavodeassi et al., 2019*; *Kim and Lemke, 2006*). The most similar outcomes were observed in temporally controlled cyclopamine treatments in *Xenopus* embryos, where ventral RPE differentiation was disrupted, but a retinal fate transformation was not reported (*Perron et al., 2003*). It therefore remains unclear if the requirement for *Gas1* during early eye and retinal development is related to Shh signaling beyond what we propose here. Direct analysis of the requirements for *Gas1* in RPCs through conditional inactivation could help to clarify this.

## Lhx2 deficiency reveals potential differences in RPC receptivity to endogenous and recombinant Shh-N

Given the importance of the co-receptors in pathway activation, the upregulation of *Gli1* in the *Lhx2* CKO explants treated with recombinant Shh-N was unexpected (*Figure 5*). One possibility is residual co-receptor activity after *Lhx2* inactivation. Indeed, *Cdon* (mRNA and protein) and *Gli1* mRNA expression persisted in the dorsal retina for a few days after tamoxifen treatment, albeit in disrupted (*Cdon*) and diminished (*Gli1*) patterns (*Figure 7BFigure 7—figure supplement 1A, C*). However, 20 nM Shh-N treatment induced *Gli1* across the CKO retina (*Figure 5E*) and although we did not track the axial orientation of the explants, this continuous pattern was consistently observed, making it unlikely that we were only sampling dorsal retina.

Another possibility is that recombinant Shh-N bypassed the requirement for co-receptors in pathway activation. Supporting this, Ptch1 and Shh-N can form ternary complexes (Ptch1:Shh-N:Ptch1) that are capable of promoting signaling although with lower efficiency compared to complexes containing

coreceptors (i.e. Cdon:Shh-N:Ptch1) (*Beachy et al., 2010*; *Qi et al., 2019*; *Qi and Li, 2020*; *Qi et al., 2018*). Another consideration is that endogenous Shh is post-translationally lipidated with cholesterol and palmitoyl moieties, key adducts for efficient signaling (*Manikowski et al., 2018*). Following secretion by Shh-producing cells, endogenous Shh is kept soluble while in transit to responding cells by forming a 'lipid-shielding' complex with SCUBE proteins (*Tukachinsky et al., 2012*). The Shh:Scube2 complex initially forms a ternary complex with Cdon or Boc on the responding cell. The Shh:Scube2 complex is then transferred to Gas1, which releases Shh from Scube2 and allows Ptch1 to bind Shh to initiate signaling (*Wierbowski et al., 2020*). In contrast, the recombinant Shh-N used here is not lipid modified and is soluble in its native form. It can directly bind Ptch1, obviating the need for Gas1 to disengage Shh-N from a complex with Scube. In the context of the *Lhx2* CKO retina, then, endogenous Shh could have been rendered less efficient than recombinant Shh-N because of the differential requirement for the co-receptors to receive endogenous Shh and remove Scube.

## Lhx2 regulates multiple pathway components to achieve the optimal level of signaling

Since *Cdon* and *Gas1* are downregulated in the retina by E14.5 but Shh signaling persists, *Lhx2* regulation of their expression is only relevant for pathway activation. However, *Lhx2* inactivation at E14.5 also reduced pathway activity. It's possible that *Lhx2* regulates another co-receptor such as *Lrp2* to sustain Shh signaling (*Figure 10B*) but this is unlikely since the *Lrp2* CKO retina does not have a phenotype at this stage (*Cases et al., 2015*). Rather, we propose that the role for *Lhx2* in sustained signaling is co-receptor independent and through regulation of other cell intrinsic pathway components. Our data does not support the existence of a single, essential pathway component that is under strong *Lhx2* regulation, but instead suggests that *Lhx2* exerts a more subtle regulation of multiple pathway components (*Figure 10B*), and we identified *Smo*, *Gli2*, and *Gli1* as candidate Lhx2-dependent genes. Genetic reductions in *Smo* or *Gli2* on their own do not exhibit haploinsufficiency (*Mo et al., 1997*; *Sakagami et al., 2009*), but their combined reduced expression could result in pathway sensitization that if strong enough, could cause a synthetic haploinsufficiency. Supporting this, *Gli2* heterozygous mice are phenotypically normal, but exhibit greater teratogenic sensitivity to vesmodegib, a small molecule Smo inhibitor, as compared to their wild type littermates (*Heyne et al., 2016*). If the reductions in *Smo* and *Gli2* expression were still not sufficient to reduce signaling, additional changes in other pathway components such as *Gli1* could have shifted the balance. Although we did not uncover evidence of Lhx2 binding in or near the *Gli1* locus, it is possible that *Lhx2* exerts some control over *Gli1* expression (*Figure 10B*). Importantly, an interaction of this nature would have to be context specific since *Gli1* inactivation, on its own, has minimal effect on development and expression of Shh target genes (*Figure 9B*; *Bai et al., 2002*; *Furimsky and Wallace, 2006*; *McNeill et al., 2012*; *Park et al., 2000*; *Wall et al., 2009*). Thus, subtle, or partial regulation of multiple pathway genes by Lhx2 could confer optimal levels of signaling, first during activation in conjunction with strong regulatory input to the coreceptors, and during sustained signaling, after co-receptor downregulation. This could be especially important in the retina where *Shh* expression is comparatively lower than in other tissues but exhibits a qualitatively similar level of *Gli1* expression (*Sigulinsky et al., 2021*). Related to this, it was interesting that there were more mCitrine + *open* loop cells in cocultures with *Lhx2* CKO explants than control after 96 hr (*Figure 4J*) indicative of an increase in ligand-mediated signaling. This difference could reflect an increase in ligand availability in the *Lhx2* CKO explants secondary to a reduction in co-receptor-dependent ligand internalization in the Lhx2-deficient RPCs or increased RGC production. It is also possible that the physical interface between the CKO retina and *open-loop* cells was more permissive to Shh signaling.

In sum, we propose that *Lhx2* regulates the expression of multiple pathway components required for optimal Shh signaling in RPCs, both during and after pathway activation. Although *Lhx2* does not regulate Shh availability at a functional level, it does limit the production of RGCs (*Gordon et al., 2013*). Because RGCs are the primary source of retinal Shh, *Lhx2* acts in an indirect but semi-autonomous manner to limit Shh expression in the retina. Furthermore, since *Lhx2* is an essential RPC transcription factor, it also likely acts to link Shh signaling to downstream targets as suggested by the persistent phenotypic severity of the *Ptch1, Lhx2* dCKO retina. Through this molecular and cellular circuitry, Shh signaling is tailored by *Lhx2* to meet the demands of early retina formation.

# Materials and methods

**Key resources table**

| Reagent type (species) or resource | Designation | Source or reference | Identifiers | Additional information |
|---|---|---|---|---|
| Gene (*Mus musculus*) | *Lhx2* | GenBank | Gene ID:16870 | |
| Genetic reagent (*Mus musculus*) | *Lhx2*<sup>flox</sup> | in lab >5 yr; from Ed Monuki, University of California, Irvine | MGI:3772179 | |
| Genetic reagent (*Mus musculus*) | *Lhx2* KO | in lab >5 yr; from Heiner Westphal, National Institutes of Health | MGI:1890208 | |
| Genetic reagent (*Mus musculus*) | *Lhx2* CKO | experimentally generated; not maintained | This paper | |
| Genetic reagent (*Mus musculus*) | *Hes1*<sup>CreERT2</sup> | in lab >5 yr; from Charles Murtaugh, University of Utah | MGI:4412375 | |
| Genetic reagent (*Mus musculus*) | *Gli1*<sup>lacz</sup> | Jax stock 8211 | MGI:2449767 | |
| Genetic reagent (*Mus musculus*) | *Ptch1*<sup>flox</sup> | from Michael Lewis, Baylor College of Medicine | MGI:2675356 | |
| Genetic reagent (*Mus musculus*) | *Ptch1* CKO | experimentally generated; not maintained | This paper | |
| Genetic reagent (*Mus musculus*) | *Rosa26*<sup>ai14</sup> | Jax stock 7914 | MGI:3809524 | |
| Genetic reagent (*Mus musculus*) | *Rosa26*<sup>mTmG</sup> | present in *Ptch1*<sup>flox</sup> strain | MGI:3716464 | |
| Cell line (*Mus musculus*) | NIH 3T3 open-loop responder cells | Pulin Li, Massachusetts Institute of Technology; *Li et al., 2018* | | |
| Antibody | Anti-Caspase 3 (Rabbit polyclonal) | BD Biosciences | PPID:AB_397274 | IF (1:500) |
| Antibody | Anti-Pou4F (Goat polyclonal) | Santa Cruz | RRID:AB_673441 | IF (1:500) |
| Antibody | Anti-Otx2 (Goat polyclonal) | Gene Tex | RRID:AB_2157172 | IF (1:600) |
| Antibody | Anti-PCNA (Mouse monoclonal) | Santa Cruz | RRID:AB_628110 | IF (1:500) |
| Antibody | Anti-Cyclin D1 (Rabbit polyclonal) | Abcam | RRID:AB_443423 | IF (1:400) |
| Antibody | Anti-b-Gal (Rabbit polyclonal) | Cappel | RRID:AB_2313707 | IF (1:5,000) |
| Antibody | Anti-Cdon (Goat polyclonal) | R&D Systems | RRID:AB_2078891 | IF (1:600) |
| Antibody | Anti-Gas1 (Goat polyclonal) | R&D Systems | RRID:AB_2107951 | IF (1:300) |
| Antibody | Anti-GFP (Chicken polyclonal) | Aves Labs | RRID:AB_10000240 | IF (1:2,000) |
| Antibody | Anti-Shh (Rat monoclonal) | DHSB | 5E1, RRID:AB_528466 | Blocking antibody (15 nM) |
| Antibody | Anti-Gli1 (Mouse monoclonal) | Cell Signalling Technologies | RRID:AB_2294746 | Western blot (1:3000) |
| Recombinant DNA reagent | pCIG-GFP | Guoqiang Gu, Vanderbilt University; *Megason and McMahon, 2002* | | Electroporation (3 µg/µl) |
| Recombinant DNA reagent | pCIG-Cdon | Ben Allen, University of Michigan; *Allen et al., 2007* | | Electroporation (3 µg/µl) |
| Recombinant DNA reagent | pCIG-Gas1 | Ben Allen, University of Michigan; *Allen et al., 2007* | | Electroporation (3 µg/µl) |
| Sequence-based reagent | Oligonucleotides | *Supplementary file 7* | | |

*Continued on next page*

*Continued*

| Reagent type (species) or resource | Designation | Source or reference | Identifiers | Additional information |
|---|---|---|---|---|
| Peptide, recombinant protein | Shh-N | R&D Systems | Cat#:1845-SH-025 | |
| Commercial assay or kit | Click-iT EdU Alexa Fluor 647 imaging kit | Thermo Fisher Scientific | Cat#:C10340 | |
| Commercial assay or kit | Cytiva Protein G HP SpinTrap Columns | Thermo Fisher Scientific | Cat#:45001485 | Affinity purification for 5E1 antibody |
| Commercial assay or kit | Amicon Ultracel-30 filter | EMD/Millipore | Cat#:UFC503024 | Buffer exchange for 5E1 antibody |
| Chemical compound, drug | Purmorphamine | EMD Biosciences | Cat#:540220 | |
| Software, algorithm | GraphPad Prism (version 9.0) | GraphPad Software, Inc | | Graphing and statistics analysis |
| Software, algorithm | Ingenuity Pathway Analysis Suite | Qiagen, Inc | | Pathway analysis for RNA seq data |
| Other | DAPI | Sigma | Cat#:D9542 | IF (300 nM) |

## Animals

All procedures and experiments involving animals were approved by the Institutional Animal Care and Use Committees at the University of Utah (protocol #11–10010) and Vanderbilt University (protocol M1500036) and set forth in the Association for Research in Vision and Ophthalmology (ARVO) Statement for the Use of Animals.

Single night matings were set up in the late afternoon and females were checked for plugs the next morning. Embryonic age determinations were based on plug date with noon designated as E0.5, by weight determinations, and by morphological criteria (*Theiler, 2013*). Pregnant dams were euthanized with a Euthanex EP-1305 $CO_2$ delivery system according to manufacturer's instructions and AALAC guidelines. Uteri were removed and embryos retrieved in Hanks Buffered Saline Solution (HBSS) supplemented with 20 mM HEPES and 6 mg/ml glucose at room temperature. The embryos were removed from the placenta and extraembryonic membranes, rapidly euthanized by decapitation with surgical scissors. Embryonic tissue was collected for PCR genotyping and whole heads, or dissected eyes were processed as needed for each analysis (see below).

## Genetics and breedings

All alleles used in this study were generated previously and are listed in Key Resources Table. *Hes1^CreERT2^*; *Lhx2^flox/+/-^*; *Rosa26^ai14^* strains are described in *Gordon et al., 2013*. The combinatorial strains needed for experiments utilizing the *Ptch1^flox^* and *Gli1^lacz^* alleles were produced through successive rounds of strategic breeding beginning with crosses to *Hes1^CreERT2^*; *Lhx2^+/-^* and/or *Lhx2^flox/flox^*; *Rosa26^ai14/ai14^* mice. To generate embryos for analysis, all breedings were done in which the male carried the *Hes1^CreERT2^* allele in a heterozygous state. Embryos that underwent Cre recombination were selected at time of dissection by detection of tdTomato or eGFP using an epi-fluorescence stereomicroscope. When possible, CKO and control embryos were segregated by the presence or absence of microphthalmia, respectively. All genotypes were verified by PCR. Primer sequences are listed in +/-. Deletion of *Ptch1* exon 3 was confirmed by an established RT-PCR protocol (*Butterfield et al., 2009*) using the same retinal RNA preparations used for qPCR. All PCR conditions are available on request.

## Tamoxifen treatments

Tamoxifen (Sigma T5648) was dissolved in corn oil (Sigma C8267) and administered by oral gavage to pregnant dams at dosages ranging between 0.15–0.2 mg per gram body weight. For *Ptch1* CKO and dCKO matings, two doses of 0.1 mg per gram body weight were administered 24 hr apart. Treatment times are noted in the text for each experiment.

## Cell lines

NIH3T3 (ATCC:CRL-1658; mycoplasma negative) was the parental cell line for the *open-loop* responder cell line used here. The generation of the *open-loop* responder line was previously described (*Li et al., 2018*). The *open-loop* responder cell line was validated for the current study by its response to recombinant Shh-N in a dose response experiment (*Figure 4*, *Figure 4—figure supplement 1*).

## Western blots

Western blots were done as described in *Ringuette et al., 2016* with the following modifications. For E15.5 samples, 10 retinas (5 embryos) per genotype were pooled and 6 retinas (3 pups) were pooled for P0 samples. A total of 50 µg protein was loaded per lane. Primary antibodies used were mouse anti-Gli1 (1:3000, Cell Signaling Technologies) and mouse anti-γ-Tubulin (1:1000, Sigma, Cat#T6557), the latter serving as the loading control. Detection was done with donkey anti-mouse IgG horseradish peroxidase at 1:5000 (1:5000, Millipore Cat#AP308P) and Luminata Crescendo Western HRP substrate (Millipore Cat#WBLUR0100). Blots were probed for Gli1 first, stripped, and reprobed for γ-Tubulin.

## Immunohistology and in situ hybridization

Embryo heads, eyes, or explants were fixed in 4% PFA/1xPBS at 4 °C from 45 min (explants) to 2 hr (heads), rinsed in PBS, cryoprotected in 20% sucrose/PBS, embedded in OCT (Sakura Finetek, Torrance, CA), and stored at –80 °C. Frozen tissues were sectioned on a Leica CM1950 cryostat at a thickness of 12 µm.

Primary antibodies are listed in Key Resources Table. Primary antibodies were followed with species-specific secondary antibodies conjugated to Alexa Fluor 488, 568, or 647 (Invitrogen/Molecular Probes, Eugene, OR). EdU was detected using Click-iT EdU Cell Proliferation kit for imaging (Thermo Fisher Scientific). Nuclei were stained with 4,6-diamidino-2-phenylindole (DAPI; Fluka). Panels showing fluorescence-based protein detection are single scan confocal images obtained with a Fluoview 1000 confocal microscope (Olympus) or Zeiss LSM710 confocal microscope equipped with 20 X objective.

In situ hybridization was performed as previously described (*Gordon et al., 2013*; *Sigulinsky et al., 2008*). Probes used in this study were digoxigenin-labeled anti-sense probes against *Atoh7*, *Cdon*, *Gas1*, *Gli1*, *Shh*, and *Vsx2*.

## RNA sequencing

Each biological replicate was composed of both retinas from a single embryo. The four replicates for each genotype were collected across three litters harvested over two consecutive days. Total RNA was isolated from flash frozen retinal tissue using QIAshredder columns and RNeasy Micro kit (Qiagen, Cat#79654 and 74004). Libraries were constructed with Illumina TruSeq RNA Library Kit V2 with poly(A) selection, and 50 cycle single end sequencing was done on the Illumina Hi-Seq 2000 platform. RNA data alignment was performed by TopHat 2 (*Trapnell et al., 2009*) on MM10 reference genome followed by gene quantification into FPKM using Cufflinks (*Trapnell et al., 2010*). Additional read count per gene was generated using HTSeq (*Anders et al., 2015*). FASTQ files are deposited at GEO repository number GSE172457.

## Bioinformatics

### RNA-seq data analysis

Differential gene expression analysis was performed with DESeq2 (version 1.21.4) (*Love et al., 2014*). Features (i.e., genes) were removed that contained 0 counts across all samples or when fewer than three samples had normalized counts greater than or equal to 20. Analyses of gene set overrepresentation and activity state prediction (ASP) was done with the Canonical Pathways (CP) tool in the Ingenuity Pathway Analysis package (IPA; Qiagen). The analysis was performed on the cohort of DEGs with an FDR cutoff of 0.001 or smaller with the Ingenuity Knowledge Base (genes only) used as the reference. Gene set overrepresentation was determined with the right-tailed Fisher's exact test with the $-\log_{10}$(p-value) of 1.3 or larger considered significant. The *gene expression ratio* is the fraction of pathway genes from the reference set (Ingenuity Knowledge Base) that were identified in the analyzed DEG set. The activation state of enriched pathways with known topology is predicted by a *Z-score of Activation* algorithm (*Krämer et al., 2014*). Derived Z-values are referred to as *Activation*

*State Prediction* (ASP) scores with positive scores predicting activated pathways and negative scores predicting inhibited pathways with 2 or greater in either direction considered significant. For visualization purposes, the cutoff for the ASP score was set at 2.5 at –2.5 in *Figure 3A*. The full list of pathways is provided in *Supplementary file 2*, Canonical Pathways worksheet.

## ChIP-seq and ATAC-seq data analysis

FASTQ files from E14 Lhx2 ChIP-seq, wildtype ATAC-seq and Lhx2 CKO ATAC-seq were obtained from GEO repository number GSE99818 (*Zibetti et al., 2019*). Sequencing adaptors were trimmed away using NGmerge (*Gaspar, 2018*). Bowtie2 was used for read alignment on the GRCm38/mm10 mouse genome (*Langmead and Salzberg, 2012*). deepTools2 was used to generate bigwig files of the sequencing data which were then visualized on the UCSC genome browser (*Ramírez et al., 2016* and https://genome.ucsc.edu/index.html). Peak calling was performed using Genrich (available at https://github.com/jsh58/Genrich; *Gaspar, 2022*). High mapping-quality reads were kept (MAPQ >10) and mitochondrial aligned reads and PCR duplicate reads were filtered out. The blacklisted regions in mouse were excluded from peak regions (https://github.com/Boyle-Lab/Blacklist/blob/master/lists/mm10-blacklist.v2.bed.gz; *Amemiya et al., 2019*; *Boyle, 2021*). ChIPseeker was used to annotate Lhx2 ChIP-seq peaks and identify the closest gene to the peak or genes within 10 kb of the peak (*Yu et al., 2015*). Lhx2-associated genes were then analyzed for gene set overrepresentation with the KEGG database as the reference, using the enrichKEGG function in clusterProfiler (*Yu et al., 2012*). For ATAC-seq peak calling, the -j option was used to set up Generich in ATAC mode. A consensus list of peaks was generated by merging all the ATAC peaks sets and filtering peaks that were reproducible in at least two samples. DESeq2 was used to identify differential accessible chromatin regions (DARs) between wildtype and Lhx2 CKO ATAC datasets from the consensus peak list (*Love et al., 2014*). Two criteria were used to associate Shh pathway genes with Lhx2 binding and DARs: ChIP-seq and ATAC-seq peaks were located within introns or 10 kb of the intergenic regions upstream or downstream of the gene body, or the assigned gene was the closest gene to the peaks.

## Quantitative reverse transcription PCR (qPCR)

Relative changes in gene expression were determined with the delta-delta-Ct method (DDCt). *Gapdh* served as the internal control gene for the initial normalization (DCt values). For E14.5 *Lhx2* CKO and control retinas, DDCt values were generated by normalizing DCt values to the mean DCt value of the control samples for each gene. In the Shh-N dose response experiments, DDCt values were generated by normalizing DCt values to the mean DCt value of the control samples at the start of the experiment (t=0). For samples from the E15.5 combinatorial *Lhx2* and *Ptch1* CKO and control retinas, E17.5 *Lhx2* CKO and control retinas, and E15.5 *Gli1* KO and control retinas, DDCt values were generated by normalization to a single control sample that was designated as a *reference*. The reference was used for all runs to control for potential batch-effect variation due to qPCR runs being done on different plates or days. Data is presented in graphs as the *fold change* in gene expression based on RQ values ($2^{-DDCt}$).

## Sybr Green assays

Sybr Green-based qPCR was done on samples from E14.5 *Lhx2* CKO and control retinas, and for the dose response experiments with recombinant Shh-N in retinal explants. Total RNA was isolated with RNeasy Micro and cDNA was synthesized with the SuperScript III First Strand Synthesis System (Invitrogen, Cat#18080–051). qPCR was done on a BioRad CFX96 Real Time PCR system with SsoAdvanced Universal SYBR Green Supermix (BioRad, Cat#170–8841) and primers detecting *Vsx2*, *Ascl1*, *Sox2*, *Lhx2*, *Pax6*, *Lin28b*, *Prtg*, *Gas1*, *Cdon*, *Boc*, *Gli*, *Shh*, *Smo*, and *Gapdh* (*Supplementary file 7*). All primer pairs were first validated for reaction efficiency and specificity.

## TaqMan assays

TaqMan-based qPCR was done on samples from E15.5 combinatorial *Lhx2* and *Ptch1* CKO and control retinas, E17.5 *Lhx2* CKO and control retinas, and E15.5 *Gli1* KO and control retinas. Total RNA was isolated using TRIzol (Thermo Fisher Scientific, Cat#15596026) and cDNAs were synthesized using SuperScript IV VILO master mix (Thermo Fisher Scientific, Cat# 11766051). qPCR was done on QuantStudio 3 Real Time PCR Systems (Thermo Fisher Scientific) with the TaqMan gene expression Master

Mix (Thermo Fisher Scientific, Cat# 444557) and TaqMan gene probes for *Gli1*, *Hhip*, *Ptch1*, *Ptch2*, *Gapdh* (**Supplementary file 7**).

### Data analysis

Data collection was done with BioRad CFX manager and ABI QuantStudio 3 package for Sybr Green- and TaqMan-based qPCR, respectively. Graphing and hypothesis testing were done with Microsoft Excel (version 16.43) and GraphPad Prism (version 9.0). Descriptive statistics for RQ values (sample number (n), mean, standard deviation (SD), standard error from mean (SEM)), hypothesis tests, testing parameters, and test results are provided in **Supplementary files 3 and 5**. All hypothesis testing was done on DDCt values.

## Cocultures to test endogenous Shh bioavailability

### Preparation and passaging of the NIH3T3 open-loop responder cell line

Cells were maintained up to 14 passages in 60 mm$^2$ petri dishes and cultured in 1 x DMEM supplemented with 10% calf serum. 8.8x10$^5$ cells were plated in 60 mm$^2$ dish for passaging (3–4 days). For experiments, 1.6x10$^5$ cells were seeded into each well of the 24 well plates and grown to >90% confluence before beginning the experiment (2–3 days).

### Dose-response testing of open loop cells

Shh-N was added to the culture medium at the doses indicated in the text, which were empirically determined in pilot experiments. A 50% medium exchange with Shh-N at the original concentration was done at 2 DIV. Once pilot experiments were completed, the dose response experiment was repeated in two separate trials and each concentration was duplicated in each trial. Image capture of microscopic fields were semi-random samplings based on two criteria: the microscopic field had to be taken in the central region of the well and the cell density was confluent, as determined by phase-contrast microscopy.

### Co-cultures

Whole retina was dissected away from other ocular tissues and placed onto the underside of a 0.4 μm Biopore insert (Millipore, PICM3050) with the apical layer of the retina in contact with the Biopore membrane (unless noted otherwise). The insert was turned over and placed into a well of a 24-well plate containing a confluent monolayer of NIH3T3 open-loop responder cells. Contact between the basal surface of the retina and the cell monolayer was achieved by removing the filter insert support legs and applying gentle manual pressure to the insert for approximately 4 s. The culture medium was raised to the bottom of the insert (approximately 300 μl) with 60–100 μl media changes per day. Once pilot experiments were completed, the co-culture experiments were run in two independent trials. Images, quantifications, and statistics are presented for the second trial. Prior to culturing, 5E1 blocking antibody was affinity purified with Cytiva Protein G HP SpinTrap Columns (Thermo Fisher Scientific 45001485) according to manufacturer's instructions. Buffer exchange from elution buffer (100 mM glycine) to sterile PBS (pH 7.4) was done with Amicon Ultracel-30 centrifugal filters (EMD/Millipore UFC503024).

### Live imaging, post-image processing, and data analysis

Wide-field epifluorescence images were taken with a Nikon DS Qi2 camera on a Nikon TE-200 inverted microscope with a 10 x objective for the dose response cultures and a 4 x objective for the cocultures. All comparable images were captured with similar camera settings and illumination. Potential daily variations in illumination and sample background fluorescence were managed with post capture background subtraction (see next paragraph).

Image processing and quantification were done with ImageJ (version 2.1.0). In brief, images were processed to obtain regions of interest (ROI) masks for identification of mCitrine + objects (nuclei). All images were first processed with background subtraction (rolling ball algorithm) and bit depth conversion from 14 to 8 prior to mask generation and measurement. Masks were then generated through thresholding, binarization, and watershedding. Masks were then applied to their respective images for object counts and pixel intensity calculations (average mean intensity per object, sum

of integrated densities sum of the products of mean pixel intensities for each object by the area of each object (mean pixels x $\mu m^2$)). Objects smaller than 90 $\mu m^2$, an empirically estimated measure of a nucleus, were excluded. Objects that passed this filter were retained for quantification and considered to represent cells based on 1 nucleus/cell and designated as such. Objects with larger areas were attributed to overlapping mCitrine + nuclei. These larger objects were also retained for analysis since their exclusion would exacerbate underrepresentation of cell counts and fluorescence intensities in the conditions with the highest number and brightest objects. Some inflation of fluorescence intensity was likely due to overlapping nuclei although the use of average mean intensities per object as a measure should have substantially reduced the impact of this confounding variable.

Graphing and hypothesis testing were done with Microsoft Excel (version 16.43) and GraphPad Prism (version 9.0). Descriptive statistics (sample number (n), mean, standard deviation (SD), standard error from mean (SEM)), hypothesis tests, testing parameters, and test results are provided in *Supplementary file 4*.

## Organotypic suspension cultures

Whole retina and lens were dissected away from other ocular tissues (explants) and placed into a 14 ml round bottomed snap cap tube with 1 ml of retina culture medium (RCM) and rotated at 15 RPM on a carousel with an axis of rotation at 30° above horizontal. Explants were incubated at 37 °C in a humidified, 5% $CO_2$ atmosphere. RCM is composed of 1 x DMEM/F12 (US Biological, Cat# D9807-05), 1% fetal bovine serum (Thermo Fisher Scientific, Cat#16140071), 6 mg/ml glucose (Sigma, Cat# G7528), 0.1% $NaHCO_3$ (Thermo Fisher Scientific, Cat#25080–094), 50 mM HEPES (Thermo Fisher Scientific, Cat# 15630–080), 1 mM glutamax (Thermo Fisher Scientific, Cat# 35050–061), 1 x N2-plus supplement (R&D Systems, Cat#AR003), and 1 x penicillin/streptomycin (Thermo Fisher Scientific, Cat# 15070–063).

At the end of the culture period, explants were rinsed in PBS. For in situ hybridization or immunohistology, explants were fixed in 4%PFA and prepared for cryostorage (see above). For qPCR, lens tissue was removed, and retinas were snap frozen in liquid $N_2$, and stored at –80 °C until use.

## Purmorphamine and Shh-N treatments

Purmorphamine (EMD Chemicals, Cat# 540220) was dissolved in 100% DMSO at 2.5 mM and stored at –20 °C. Explants were treated with 10 µM purmorphamine or vehicle (0.4% DMSO). *E. coli*-derived human N-terminal modified (C24II) fragment of Sonic Hedgehog (Shh-N; R&D Systems, Cat#1845-SH/CF) was reconstituted in PBS at 10 µM and stored at –80 °C. Explants were treated with Shh-N at the concentrations indicated. When possible, multiple breeding pairs were set per single night mating to increase sample size. Explants were cultured for 24 hr.

See qPCR methods for data collection and analysis.

## Ex vivo electroporation

*Lhx2* CKO, *Gli1*^lacz/+ embryos were identified by phenotyping and rapid PCR genotyping. Explants were transferred into sterile PBS without $Ca^{2+}$ and $Mg^{2+}$. A total of 1.5 µl of DNA (3 µg/µl) in 30% glycerol with methyl green was pipetted onto the apical retinal surface, and electroporated with a BTX ECM830 (5x50ms pulses at 50 V, 250ms intervals). One explant per animal received the GFP control plasmid (*pCIG*) and the other received an equimolar 50:50 mixture of *pCIG-Cdon* and *pCIG-Gas1*. Explants were transferred to 1 ml of RCM and cultured. 3 nM Shh-N and 1 µM EdU were added after 24 hr, and the cultures were maintained for an additional 24 hr. Explants were fixed for 1 hr in 4% PFA and cryopreserved until use.

Sectioned explants were stained for GFP to identify transfected cells, EdU to identify proliferating cells (RPCs), and β-Gal to identify *Gli1* expressing cells. Cells were first scored for the co-localization of GFP and EdU. Once completed, GFP, EdU double positive cells were scored for β-Gal expression. Counting was done on at least three sections per explant, and when possible, at least 100 GFP, EdU double positive cells were counted per section. Graphing and hypothesis testing were done with Microsoft Excel (version 16.43) and GraphPad Prism (version 9.0). Descriptive statistics (counts per explant, sample number (n), mean, standard deviation (SD), standard error from mean (SEM)), hypothesis tests, testing parameters, and test results are provided in *Supplementary file 6*.

## Materials availability statement

Requests for *open-loop* cells should be directed to Dr. Pulin Li. All other requests for materials not commercially available should be directed to Dr. Edward Levine.

## Acknowledgements

We thank Michael Lewis (Baylor College of Medicine, Houston TX) for the *Ptch1*^flox^; *Rosa26*^mT/mG^ mice, Guoqiang Gu (Vanderbilt University) for the *pCIG-GFP* plasmid, Ben Allen for the *pCIG-Cdon* and *pCIG-Gas1* plasmids, Joe Brzezinski for training on the electroporation technique, JP Cartailler (Creative Data Solutions) for bioinformatics. We also thank members of the Levine laboratory at the University of Utah (Anna Clark, Crystal Sigulinsky) and Vanderbilt University Medical Center (Allison Klinger, Dianna Rowe, Mahesh Rao, Amanda Leung,) for their technical and intellectual support. We also thank Sabine Fuhrmann and the Fuhrmann laboratory for sharing reagents, technical expertise, and intellectual input. This work was supported by funds from the National Institutes of Health to EML (NEI R01-EY013760, NEI P30-EY014800), SB (NEI RO1-EY020560), PL (NICHD R00-HD087532), AF (NEI T32-EY007315), and PG (NICHD T32-007491). Additional funding was generously provided through NEI P30-EY008126 and by unrestricted grants to the John A Moran Eye Center (University of Utah) and Vanderbilt Eye Institute (Vanderbilt University Medical Center) from The Research to Prevent Blindness, Inc.

## Additional information

### Funding

| Funder | Grant reference number | Author |
|---|---|---|
| National Eye Institute | R01-EY013760 | Edward M Levine |
| National Eye Institute | P30-EY014800 | Edward M Levine |
| National Eye Institute | P30-EY008216 | Edward M Levine |
| National Eye Institute | R01-EY020560 | Seth Blackshaw |
| National Eye Institute | T32-EY007315 | Alexandra W Fuller |
| Eunice Kennedy Shriver National Institute of Child Health and Human Development | T32-007491 | Patrick J Gordon |
| Eunice Kennedy Shriver National Institute of Child Health and Human Development | R00-HD087532 | Pulin Li |
| Research to Prevent Blindness | | Edward M Levine |

The funders had no role in study design, data collection and interpretation, or the decision to submit the work for publication.

### Author contributions

Xiaodong Li, Alexandra W Fuller, Formal analysis, Validation, Investigation, Visualization, Writing – review and editing; Patrick J Gordon, Conceptualization, Data curation, Formal analysis, Validation, Investigation, Visualization, Methodology, Writing – original draft, Writing – review and editing; John A Gaynes, Formal analysis, Validation, Investigation, Visualization, Methodology, Writing – review and editing; Randy Ringuette, Investigation, Visualization; Clayton P Santiago, Data curation, Visualization, Writing – review and editing; Valerie Wallace, Supervision, Funding acquisition, Writing – review and editing; Seth Blackshaw, Resources, Supervision, Funding acquisition, Visualization, Writing – review and editing; Pulin Li, Resources, Funding acquisition, Methodology, Writing – review and editing; Edward M Levine, Conceptualization, Formal analysis, Supervision, Funding acquisition, Investigation,

Visualization, Methodology, Writing – original draft, Project administration, Writing – review and editing

### Author ORCIDs
Xiaodong Li http://orcid.org/0000-0003-3328-3580
Clayton P Santiago http://orcid.org/0000-0001-7191-668X
Valerie Wallace http://orcid.org/0000-0003-3721-9017
Seth Blackshaw http://orcid.org/0000-0002-1338-8476
Edward M Levine http://orcid.org/0000-0003-1725-2805

### Ethics
All procedures and experiments involving animals were approved by the Institutional Animal Care and Use Committees at the University of Utah (14-09013) and Vanderbilt University (M1500036) and set forth in the Association for Research in Vision and Ophthalmology (ARVO) Statement for the Use of Animals.

### Decision letter and Author response
Decision letter https://doi.org/10.7554/eLife.78342.sa1
Author response https://doi.org/10.7554/eLife.78342.sa2

---

## Additional files

### Supplementary files
- Supplementary file 1. Differential gene expression analysis of RNA-seq data with DESeq2.
- Supplementary file 2. KEGG pathways identified by Ingenuity Pathway Analysis and clusterProfiler.
- Supplementary file 3. Statistics and tests for multiple pairwise comparisons of qPCR-based measurements of gene expression.
- Supplementary file 4. Statistics and test results for experiments with open-loop responder cells.
- Supplementary file 5. Statistics and test results for Shh-N dose responses in retinal explants.
- Supplementary file 6. Statistics and test results for cell counts following electroporation in *Lhx2* CKO, *Gli1*$^{Lacz/+}$ retinal explants.
- Supplementary file 7. Oligonucleotides.
- MDAR checklist

### Data availability
Sequencing data have been deposited in GEO under accession code GSE172457. All data generated or analyzed are included in the manuscript, supporting files, and supporting tables.

The following dataset was generated:

| Author(s) | Year | Dataset title | Dataset URL | Database and Identifier |
|---|---|---|---|---|
| Levine EM, Gordon PJ | 2021 | RNA sequencing of E15.5 retinal tissue from control and Lhx2 CKO mice | https://www.ncbi.nlm.nih.gov/geo/query/acc.cgi?acc=GSE172457 | NCBI Gene Expression Omnibus, GSE172457 |

The following previously published dataset was used:

| Author(s) | Year | Dataset title | Dataset URL | Database and Identifier |
|---|---|---|---|---|
| Zibetti C, Liu S, Wan J, Qian J, Blackshaw S | 2017 | Lhx2 controls changes in chromatin accessibility in retinal progenitor cells during developmental competence transitions | https://www.ncbi.nlm.nih.gov/geo/query/acc.cgi?acc=GSE99818 | NCBI Gene Expression Omnibus, GSE99818 |

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
