## [Editor Report]

This study dissects the complex Shh pathway to explain the phenotypic similarity between Lhx2 and Shh retinal knock-out mice. The authors use multiple converging experimental strategies to show Lhx2 activates the Shh pathway, mainly by up-regulating co-receptors Gas1 and Cdon in retinal progenitor cells. The experiments are creative, and the findings provide evidence that Lhx2 acts in a contextual manner and integrates signalling pathways, conferring enhanced Retinal Progenitor Cells with the competence to respond to Shh. The study provides novel and interesting views on retinal development.

---

## [Decision Letter]

[Editors' note: this paper was reviewed by Review Commons.]

**Decision letter after peer review:**

Thank you for submitting your article "Lhx2 is a progenitor-intrinsic modulator of Sonic Hedgehog signaling during early retinal neurogenesis" for consideration by eLife. Your article has been reviewed by 2 peer reviewers at Review Commons along with an additional peer reviewer at eLife, and the evaluation has been overseen by a Reviewing Editor and Claude Desplan as the Senior Editor. The following individuals involved in review of your submission have agreed to reveal their identity: Tom Glaser (Reviewer #3).

Based on your manuscript, the reviews and your responses, we invite you to submit a revised version incorporating the revisions as outlined in your response to the reviews.

Please also address all the main point raised by the additional reviewer of your manuscript as they will help increasing the clarity of your message. The additional comments points to additional details of the manuscript that need your attention.

*Reviewer #3:*

The Lhx2 LIM homeodomain transcription factor plays multiple key roles in vertebrate eye development - specifying regions of the eye field, patterning the optic cup, maintaining the pool of mitotic retinal progenitor cells (RPC), modulating retinal cell fate determination (histogenesis), and integrating signaling pathways. In germline Lhx2 mutants, morphogenesis arrests at the optic vesicle stage.

All paracrine signals have two parts - the sending and responding cells. In this study, Li, Gordon et al. explore how neural progenitors acquire the competence to receive and transduce sonic hedgehog signals - via expression of coreceptors and other pathway components, regulated in large part by Lhx2.

They deeply explore the mechanism of Lhx2 action in RPCs. They offer plausible explanations for the phenotypic similarity between Lhx2 and Shh retinal CKO (conditional knockout) mice. To test their central hypothesis - that Lhx2 confers RPCs with enhanced competence to respond to Shh signaling by positively regulating expression of Gas1 and Cdon coreceptors - they push and pull the Shh pathway at multiple levels. They employ informative mutants, Hes1-CreERT temporal loss-of-function mice (to delete Lhx2 in all RPCs at the onset of histogenesis, from embryonic day E11.5), pharmacological treatment (N-Shh, PMA agonist), bypass experiments, and creative bioassays (e. g. retinal explant cocultures with apposed open-loop NIH3T3 fluorescent Shh reporter cells). They assess molecular effects of Lhx2 deletion on mRNA (bulk RNA-seq, qPCR) and protein abundance, chromatin accessibility, and informative reporter expression (e. g. Gli1-lacZ knock-in allele) and evaluate Lhx2 genomic targets (ChIP-seq). The various data converge. The authors apply standard pathway logic (molecular epistasis) to define the major steps and general mechanisms through which Lhx2 modulates Shh signaling.

The results are complicated and do not fully explain all phenotypes. Lhx2 is neither necessary nor sufficient for Shh action. Signaling is attenuated but not eliminated in mutant RPCs. Despite this complexity, the study advances understanding and provides a basis to explore how signaling pathways are integrated and finely tuned.

1. The initial data are presented in a confusing order. After noting similarity between Lhx2 and Shh retinal CKO phenotypes, the authors show in situ hybridization images (Fig 1) suggesting that Shh signaling (Gli1) - but not Shh ligand - is downregulated in Lhx2 CKO retina. This is the major premise for the study - and title of the first Results section. However, the ISH data are not quantitative (or persuasive), and the CKO histological phenotype does not obviously differ from control at E14 ('cell type profiles' in line 132, a bit vague), so the manuscript does not start with a strong foundation. However, the Gli1 and Shh qPCR data (Fig 6B) are convincing and do support ISH findings - and should be presented early, before the Gli1 immunoblot (Fig 1C). Most readers will look for these quantitative mRNA data in Fig 1. They belong here, logically, since Gli1 is a well-established, sensitive readout of Shh signaling.

Likewise, the Lhx2 RNA-seq data in Fig. 2C-E are confusing. Knowing the structure of the conditional allele in advance, the authors should quantify mRNA isoform reads with WT (ex 1/2, 2/3 or 3/4 junctions), deleted (ex 1/2 junction) or unknown (reads lacking ex 1-3) splicing patterns separately, in their existing dataset. Rather than the simple schema in Fig 1E, they should show 6K WT and CKO reads aligned to the mm10 reference genome, along with the AUG and qPCR amplicon, and note whether deleted isoforms are predicted to undergo NMD and should thus have lower mRNA abundance a priori (in Methods or legend).

How much is Lhx2 mRNA downregulated within 48 hr of Tam treatment (line 130)? What are 'other' features in Fig 2B?

2. The bioinformatic rationale for exploring Lhx2 regulation of Shh pathway genes (Fig 3) is logical but presented in a jumbled way. Two orthogonal strategies are applied (three, including ATAC-seq) - and only one pathway (Hh) emerged with statistical significance in both. One approach (RNA-seq of dissected CKO retinas, with KEGG pathway topology analysis of DEGs) tests altered mRNA expression - direct and indirect regulation - whereas the ChiP-seq data (from flow-sorted wild-type RPCs in an earlier study, analyzed via clusterProfiler) tests Lhx2 chromatin occupancy - expected for direct regulation. The logic for this overlapping strategy is buried in the text, and obscured by interspersed panels in Fig 3, forcing the reader to carefully dissect the legend to understand what kind of (highly derived) data are presented. Label panels 3AB better so these methods are obvious.

3. The appositional explant assay is a compelling functional measure of bioactive Shh (Fig 5). Likewise, the Lhx2 and Ptch1 CKO and dCKO data (Fig 7) are creative and instructive, showing that Lhx2 acts upstream but that Shh signaling remains intact (attenuated) in Lhx2 CKO RPCs.

However, the qPCR normalization (Fig 7A) does not make sense (to me) - each control value should be set to 1.0 +/- SE (or SD). There is a similar normalization concern in Fig 10 (lines 701 ff notwithstanding).

Does the increased abundance of Cyclin D1+ cells (Fig 7) reflect an increase in bioactive Shh mitogen (Fig 5B)?

---

## [Author Response]

Reviewer 1: Figure 1 – Provide immunohistology of Lhx2 CKO phenotype at E15.5 in Figure 1, notably RGC and RPC markers.

Immunohistology data is now provided in Figure 1C. These data show that while the size of the retina is reduced, the RPC population is still abundant in the CKO and Lhx2 is largely absent in the CKO retina. In addition, in Supplemental Figure 1, we provide images of the eye from E15.5 control and CKO embryos, again demonstrating the smaller eye in the CKO. Additional immunohistology is also provided to show that retinal organization is largely intact, but at least one protein, the RPC marker Hes1, is reduced in its expression.

Reviewer 1: Figure 4 – Figure 4 presents a published approach and thus can be included in Figure 5.

We followed this reviewer’s advice and placed the bulk of Figure 4 into the revised Supplemental Figure 4. Certain elements of the original Supplemental Figure 4 were moved into the main figure at the suggestion of the third reviewer. In this revised version, Figure 4 and Supplemental Figure 4 now encompass what was in the original Figures 4 and 5 and Supplemental Figure 4.

Reviewer 1: Figure 5 – Explain how results were normalized based on retinal size.

This comment relates to the data in Figure 5D. We initially considered normalizing the mCitrine+ cell counts by retinal area but ultimately decided to normalize to the number of retinal explants per well (1 in all cases), arriving at the measure of *reporter+ cells/explant*. We believe this measure yields an accurate assessment of ligand availability, but we acknowledge that it doesn’t account for the potential variability in the amount of retinal tissue in each culture. Therefore, we went back and calculated the area of each explant and normalized the reporter+ cell counts to these values using the following measure: *number of reporter+ cells/mm^2^ of retinal explant*. We then performed 2-way ANOVA, followed by Tukey’s multiple comparisons test. When compared to the statistics of the measurements of *reporter+ cells/explant*, we generally obtained smaller p-values. Notably, two comparisons dropped below the 0.05 α value threshold: the CKO+5E1 condition versus the control condition at 72hr and 96hr. These outcomes lend even stronger support to our conclusions. We are happy to replace the existing data and statistics in Figure 5D pending agreement from the reviewer and/or editor. For now, the data is included as an additional Supplementary files for reviewer/editor consideration (Supplemental Figure 10; Supplemental Table 11).

This change is now incorporated into the revised manuscript.

Reviewer 1: original Figure 8 (now Figure 9) – Lhx2 staining should be done to confirm that the reporter+ cells are not escapers.

We addressed the valid concern of potential escaper cells (Lhx2+ cells in the CKO retina) by performing electroporations with the pCIG plasmid (GFP expression only) into control and *Lhx2* CKO retinal explants (the original plan highlighted above). GFP+ cells were scored for co-expression of Lhx2 and b-Gal. As shown in Figure 8F, there were very few cells that co-expressed Lhx2 and b-Gal in the CKO retina, even though b-Gal could be detected in CKO explants electroporated with pCIG alone. The presence of b-Gal+ cells is not surprising since the culture medium contained Shh-N (3nM). This is the likely explanation for the b-Gal+ cells in the pCIG-electroporated CKO explants that were served as the control condition for the coreceptor overexpression experiment (Figure 8E, black bar; representative images are provided Figure 8-Supplementary Figure 1E). This data further supports our position that the increased proportion of b-Gal+ cells in the CKO retinal explants transfected with Cdon and Gas1 was due to Cdon and Gas1 overexpression and not persistence of Lhx2+ cells.

Reviewer 1: additional comments – The ﬁnding that a Lhx2 controls several components of the HH pathway could be relevant to Lhx2 activity in patterning of the cortex – I suggest to discuss the possible relevance of the ﬁndings to other organs.

In the introduction, we now provide a brief description of the multifunctional nature of Lhx2 in other organ systems, notably limb and hair follicle and refer the reader to the cerebral cortex as well as to a recent review of Lhx2 in multiple developmental contexts, to provide a stronger rationale for studying Lhx2 in different developmental contexts, including the retina. In the first Discussion section, we also contrast the difference in how Lhx2 regulates Shh signaling in the limb bud, illustrating the point that a multifunctional regulator like Lhx2 can influence the same pathway, but through a different mechanism in another context.

Reviewer 1: Figure 1 – Demonstration of loss of Lhx2 protein following Cre recombination.

We showed previously that Lhx2 protein is not detected by immunohistology in most RPCs within 48 hr of tamoxifen treatment (Figure 5X and Y in Gordon et al. (2013)). This is briefly mentioned in the first paragraph of the Results section.

Reviewer 1: Figure 2 – Precisely define the biological replicates for RNA seq.

Each biological replicate for RNA seq was composed of both retinas from a single embryo. The four replicates for each genotype were collected across three litters harvested over two consecutive days. This is now stated in the Methods, subsection: RNA sequencing

Reviewer 1: Figure 4E – Provide explanation of the quantitative analysis

We made the following modifications to the manuscript to provide further clarification:

The second paragraph of the Results section “Lhx2 does not regulate Shh ligand availability”: Dose-dependent increases in mCitrine expression were observed when monitored at 24 hr intervals (Figure 4D). This observation was supported by quantification of the accumulation of mCitrine+ nuclei and the changes in fluorescence intensity over the time course of the experiment (Figure 4E) and by sum of the mean fluorescence intensities as a function of area at 72 hr (Figure 4F; Supplemental Table 4).

At the end of the same paragraph: “Taken together, these data indicate that mCitrine expression in the open loop cells are dependent on Shh-N in a dose dependent manner.”

Added to the Materials and methods subsection “Cocultures to test endogenous ligand availability”→ “Dose-response testing of open loop cells”: Once the pilot experiments were completed, the dose response experiments were performed twice with similar results. Each experiment had two wells per condition. Image capture of microscopic fields were semi-random samplings based on two criteria: the microscopic field had to be taken in the central region of the well and the cell density was confluent, as determined by phase-contrast microscopy.”

Reviewer 1: Figure 5 – Experimental details: how many independent experiments; how many retinas were tested; were retinas from the same mouse considered ‘independent’.

Additional information is provided in the methods section titled “Cocultures to test endogenous ligand availability” in the subsections: *Dose-response testing open loop cells* and *Co-cultures,* and in Supplemental Table 4, and summarized here for the co-cultures. Once conditions and measurement parameters were established in pilot experiments, the complete time course was run on two separate experimental days. As described in the results, 10 control, 13 CKO, and 2 CKO+5E1 retinas were cultured from these two experiments. Only the second experiment was used for image display, quantification, and statistics because of a partial obstruction in the illumination path of the microscope during the first experiment. Supplemental Table 4 provides the statistics from the second experiment day. Retinas from the same embryo were treated as independent samples and the 2 retinas for the CKO+5E1 condition were from separate embryos and is now indicated in Supplemental Table 4.

Reviewer 1: Supplemental Figure 7B,C – Explain how the deletion of Ptch1 was examined.

Butterfield et al. (Butterfield et al., 2009) demonstrated that deletion of the *Ptch1^flox^* allele resulted in a truncated transcript lacking exon 3 that could be detected by RT-PCR and distinguished from the unrecombined transcript by its smaller size. We used this approach to detect both the unrecombined and recombined transcripts. This is described in the second paragraph of the Results section *“Ptch1 inactivation stimulates Shh signaling in the Lhx2-deficient retina but fails to restore retinal development”* and in the last paragraph of the material and methods section “*Genetics and breedings”.* The Butterfield citation is now in both locations.

Reviewer 1: Figure 7B,C – Add control images.

Control (wild type retina) images were already provided for Figure 7C and E (1^st^ panels in each row). Control images for Figure 7B are now provided in supplemental Figure 7D.

Reviewer 1: Figure 8A,B – Indicate genotype and Figures 8A and B should be presented in one panel and same orientation.

The genotypes for the images in Figure 8A and B are now indicated in the figure. Combining Figures 8A and B into a single panel could be confusing because 8A is a developmental comparison across ages in the *Gli1^Lacz/+^* retina (wild type) and 8B is a comparison between the *Lhx2* control and CKO at a single age. Using the same orientation for 8A and 8B brings in a significant amount of empty space. We therefore split Figure 8 into two figures with Figure 8 containing the panels of A and B in the same orientation and the new Figure 9 containing the electroporation data. The original figures 9 and 10 are now 10 and 11. Text and figure legends have been adjusted.

Reviewer 1: Original Figure 8D (Now Figure 9) – Present the different channels, in addition to the merged images

We believe the reviewer is referring to panels I, Ii, J, Ji. The single channel images are provided in a new Supplemental Figure 9.

Additional changes to the new Figure 9: The labels “GFP” and “Cdon/Gas1” in the original Figure 8E-J are now replaced with pCIG and pCIG-Cdon/Gas1, respectively in Figure 9C-E.

Reviewer 2: Provide reference for Intro statement: “pathway activation is not achieved by simple binding to Patch but requires one of three coreceptors..”

The following references have been added: (Allen et al., 2011; Izzi et al., 2011)

Reviewer 2: Correct the misidentification of Notch as ligand and Dll1 and 3 as receptors in Intro.

This is now corrected

Reviewer 2: Typo in Supplementary Table 4 – 3^rd^ page – “centrations at 72 hr”

The current version should conform.

Reviewer 2: Supplementary Figure 8 mislabeled as 7.

*T*his is now corrected.

Reviewer 2: A logical explanation of Tamoxifen administration at a window E11.5 – E15.5 is good to have in the Results section.

This is now added in the first paragraph of the Results section

Reviewer 2 – Figure 2A and Result 2: Why harvesting at E14.5 speciﬁcally for qPCR/ChIP sequencing, while for RNA sequencing and in situ, it was done E15.5.

The in situ hybridizations were the first analysis done for this study and provided the rationale for RNA sequencing at E15.5. The RNA sequencing revealed robust changes in expression, so we went to E14.5 for qPCR to determine if the changes could be detected sooner. This was helpful in that it allowed us to start the cultures closer to the time of tamoxifen treatment. The E14.5 timepoint for ChIP-seq and ATAC-seq was fortuitous; it was part of another study.

Reviewer 2 – Figure 5D,E: It is not clear why there were more mCitrine+ cells in CKO explants at 96 hours.

We can only speculate on this interesting outcome. It could be that the CKO retina has a higher ligand bioavailability because the coreceptor-deficient RPCs were not internalizing ligand. Another reason could be the shift toward increased RGC production resulting in more ligand. A more speculative possibility is that extracellular environment of the CKO retina enhanced the interaction with the *open-loop* responder cells was more permissive. Importantly, it supports our conclusion that endogenous Shh was available and biologically active in the *Lhx2* CKO retina. We provide this explanation in the discussion, section *“Lhx2 regulates multiple pathway components to achieve the optimal level of signaling*”, end of first paragraph.

Reviewer 2 – Figure 6 C,D,E,F: Does Lhx2 CKO, cause cell death as the levels of vsx2 are low in vehicle in CKO (D,F) as compared to ctrl (C,E).

In Supplemental Figure 6, we show that cell death is enhanced in the RGC layer but not in the neuroblast layer where the RPCs are located, the cells that normally express Vsx2. The cell death pattern is observed in both wild type (control) and Lhx2 CKO retinas, and, as we describe in the results, is a known issue in embryonic retinal explant culture when cultured in suspension and lacking the addition of neurotrophic factors and cell death inhibitors.

Reviewer 2 – Figure 6: The levels of Gli1 are higher in CKO vehicle (D,F) than the control panel (C,E). This difference in Gli1 expression is more evident D versus C. Does it mean that CKO increases Gli1 expression in explants, which seems to be opposite of what shown in the ﬁrst results (Figure 1D, E).

The method used here for in situ hybridization is not sensitive enough to conclude that the expression levels of *Gli1* are different between the vehicle treated

Ctrl and CKO tissues. The method is suitable, however, for the robust, qualitative changes we observed for each treatment within a genotype. A better evaluation of a difference in *Gli1* expression in the CKO versus Ctrl is provided by qPCR in Figure 6G (0 nM SHH-N, 24hr timepoint), which shows the *Gli1* is lower in the CKO. This data is supported by statistical analysis (Supplemental Table 5 – Figure 6G – Tukey’s multiple comparisons test p<0.0001 for 0nM CKO vs. 0nM CTL). In this regard, all our data are consistent with reduced Gli1 expression in the *Lhx2* CKO retina when no additional treatments are done. The reason why the vehicle treated Ctrl retinal explant has reduced *Gli1* expression after 24hr is because the suspension cultures and basal medium do not support continued Shh signaling without addition of ligand or agonist. This deficiency is what allows us to use the wild type retina as a positive control for purmorphamine and SHH-N activities in these experiments.

Reviewer 2 – Figure 6: It is not clear why the increase in Gli1 expression with Shh-N (E,F) is not that much evident than with purmorphamine (C,D) in both control and CKO explants.

The SHH-N experiments and stainings were performed after the purmorphamine experiments, so each treatment should be considered independently of one another. Even if done together, it would be very difficult to compare the efficacy of SHH-N vs purmorphamine in this manner because their mechanisms of action are different: SHH-N acts through Ptch1 and purmorphamine bypasses Ptch1 and acts directly on Smo. Rather, our intent here was to ask: Can Shh signaling be stimulated in the CKO retinas with purmorphamine, and if so, with SHH-N?

Reviewer 2 – Figure 7 7-F: The changes in the levels of Cyclin D1 and Hes1 in Ctrl/Lhx2 CKO/ dCKO are not very clear from the data in present form. It would be better to show the changes in their mRNA levels by qPCR analysis. Further, it is not clear how the changes in CyclinD1 and Hes1 levels are proving that Lhx2 acts downstream of Shh signaling.

Cyclin D1 and Hes1 expression levels in RPCs are promoted by Shh signaling (Sakagami et al., 2009; Wall et al., 2009; Wang et al., 2005). Since the qPCR data in Figure 7A indicated increased Shh signaling in the dCKO, we anticipated some improvement in the phenotype compared to the *Lhx2* CKO but this was not observed (Figure 7B – E). Therefore, we propose that Lhx2 is also acting downstream of Shh signaling to drive retinal development, and this is in line with Lhx2 acting as a central RPC transcription factor. During our analysis, we observed an increase in Hes1 staining intensity in the dCKO and the graph in Figure 7F documents this. This suggests that Shh signaling, while not sufficient to restore the number of Hes1+ cells, could still be stimulating Hes1 expression in the RPCs that persist. We observed a similar effect with Cyclin D1, but in this case, the increased staining intensity was correlated with *Lhx2* inactivation (Figure 7D). Thus, at least one downstream target of Shh signaling (Hes1) supports the qPCR data in Figure 7A that *Ptch1* inactivation promoted Shh signaling in the absence of Lhx2 even though it wasn’t sufficient to improve retinal development.

With respect to Hes1 and Cyclin D1 mRNA levels, the RNA sequencing data shows them to be highly downregulated in the Lhx2 CKO (Supplemental Table 1), consistent with fewer expressing cells. In this light, we don’t believe the intensity increases observed by antibody staining in Figures 7C-E will be detected by qPCR, even if the changes in protein levels are tied to increased mRNA in the few cells that express Cyclin D1 and Hes1.

Description of analyses that authors prefer not to carry outReviewer 1: related to Figure 2 – Perform GSEA analysis for the hedgehog pathway genes.

In the first submission, we provided a gene enrichment analysis of the RNA seq data using the Ingenuity Pathway Analysis (IPA) package from Qiagen, specifically the *Canonical Pathways* analysis. While IPA and GSEA use different statistical methods to identify gene set enrichments, both provide similar information and query overlapping gene set databases (e.g. GO, KEGG) although IPA also queries an in-house, actively curated information hub (Ingenuity Knowledge Base). An additional and useful feature of *Canonical Pathways* is its pathway topology feature, yielding a Z-score to predict the activity state of the signaling pathways. In the case of the Sonic Hedgehog pathway, the Z-score predicted *inhibition* in the Lhx2 CKO retina, consistent with our other data.

Perhaps more importantly, because our study deeply investigates the connection between Lhx2 and Shh pathway, our conclusion is not strongly dependent on the gene set enrichment to identify the Shh pathway as a pathway of interest, although we were encouraged by this observation. Our primary purpose for the RNA sequencing was to efficiently identify Shh pathway genes as candidate targets of Lhx2 based on their differential expression and their associated predicted effects on signaling. For these reasons, we don’t believe that GSEA will add additional information to the study than was already obtained with IPA.

Reviewer 2: Figure 1C – Western blots, tubulin is less in CKO alongside Gli1. It would be better to do quantification for showing any significant changes.

Although this was listed in the Minor comments section, we regretfully are unable to provide a quantification without repeating the westerns because we no longer have the original scan data. We are confident in the accuracy of the western blot data as it is consistent with all other measures of Gli1 expression (in situ hybridization, RNA sequencing, qPCR, *Gli1^LacZ^* reporter), but it can be removed if deemed necessary. Its removal shouldn’t weaken the findings of the study given the abundance of other evidence showing reduced Gli1 expression in the *Lhx2* CKO.

Reviewer 2: “Authors do not seem to have performed a rescue experiment with Lhx2 CKO. If this is absolutely not possible, a conditional overexpression could have given conﬁrmatory clues on the Lhx CKO phenotype discussed. In any case some sort of rescue experiments are essential with respect to Lhx2 as done with Cdon and Gas1.”

We believe 2 questions are being asked by the reviewer’s suggestion to re-express Lhx2 following *Lhx2* inactivation: Are the changes in retinal development caused by *Lhx2* reversible? And if so, for how long? These are indeed interesting questions but addressing them would require a full study to assess cellular phenotypes, chromatin, or effects on signaling. It’s also unclear how this line of inquiry will provide further resolution to the questions asked in this study: Does Lhx2 interact with the Shh pathway, and if so, how? Even if an argument could be made that this line of inquiry is relevant to the current study, there are significant challenges with feasibility. Electroporation and the ex vivo culture platform has technical limitations for assessing cell and tissue level phenotypes over extended intervals, and while an inducible *Lhx2* transgenic mouse is published (Z/Lhx2-GFP; (Hägglund et al., 2011)), this transgene, like the *Lhx2^flox^* allele, is Cre-dependent. We therefore don’t have the ability to generate temporal windows of Lhx2 deficiency since the *Lhx2^flox^* allele and the *Z/Lhx2-GFP* transgene would be recombined at the same time.

Reviewer 2: Figure 6 – Gli1 readout at higher ShhN concentrations would shed light on the roles of co-receptors.

We believe the reviewer is suggesting the following: if the *Gli1* response in the *Lhx2* CKO retina remains attenuated at a high Shh-N concentration (20 nM, Figure 6F), then this would support the idea that the role of the co-receptors is to lower the ‘effective concentration’ of ligand needed to activate the pathway. This might be true, but we also point out that an equivalent response to 20 nM Shh-N between control and *Lhx2* CKO retinas would not negate the co-receptor model because the high ligand concentration could override the mechanisms that function at physiological concentrations. That was the basis for the dose response (Suppl Figure 6C) − To identify a concentration range that approximates physiological conditions and to use this range to make quantitative measurements (Figure 6G).

The following responses refer to the coreceptor overexpression experimentsReviewer 1: Gas1 is confined to the peripheral RPCs and ciliary margin. Explain why Cdon and Gas1 were co-electroporated and the outcome of electroporating only Cdon.

We agree that electroporating the coreceptors individually is sensible and could shed light on potential differences on how the coreceptors are utilized for Shh signaling. This was not our question, however, and we had to consider the factors described below, which ultimately led us to combined coreceptor electroporation.

– The frequency of generating an *Lhx2* CKO embryo with the *Gli1^LacZ^* reporter is 1 in 8. This very low yield required us to generate a high number of embryos for the control electroporation (GFP alone) and the combined coreceptor test condition. Adding in the single coreceptor overexpression conditions would have tripled the number of mice needed, adding considerable effort and cost to the experiment. This was difficult to justify for the intended goal of the experiment, which is described in the next bullet point.

–The intent of this experiment was to address the following question: Is the pathway downstream of the co-receptors sufficiently intact to support coreceptor-mediated activation? If true, this would add another piece of evidence that Lhx2 mediates at least some of its effect on the pathway by promoting co-receptor expression. Combining the coreceptors was the most efficient way to test this.

– In the discussion, we describe a recent study showing that Cdon and Gas1 can work cooperatively (Wierbowski et al., 2020). Considering this, it’s possible that single coreceptor electroporation would not stimulate expression of the *Gli1^LacZ^* reporter. Importantly, this wouldn’t change our stated conclusion. It could suggest that Gas1 has a role in central RPCs, and based on our expression data, at a lower level of expression than in peripheral RPCs and ciliary margin. This would be interesting, but would require significant follow up experiments, which would extend the work beyond the focus of the interaction between Lhx2 and the Shh pathway.

– Addressing coreceptor requirements is best done by loss of function. Cdon was already shown to be required for retinal Shh signaling (Kahn et al., 2017). The requirement for Gas1 in this context is not known because the earlier phenotype in the germline *Gas1* KO precludes this model from being used for this purpose. To our knowledge, this has not been examined with an inducible *Gas1^flox^* allele. These points were stated in the original submission and found in the discussion.

Reviewer 1: The outcome of electroporating the coreceptors in the control retina should be presented.

We assume that the reviewer is referring to the wild type/*Lhx2* heterozygous retina as the control retina. For the following reasons, we believe that performing these experiments in the wild type context is not easily interpretable and that the wild type retina does not meet the definition of a control in this set of experiments.

– In the control (wild type/*Lhx2* het) retina, Shh signaling is already active and the co-receptors are downregulated by E13.5 (Figure 8A, Suppl. Figure 8). This rapid downregulation of coreceptor expression following Shh pathway activation is also observed in other tissues and is interpreted to mean that coreceptors are not required for maintained signaling (Lee et al., 2001; Tenzen et al., 2006). Thus, the context of the wild type retina is different than the *Lhx2* CKO retina at the start of the culture period. Because of this, the wild type does not serve as a control in these experiments. Rather, the empty vector (GFP only) in the *Lhx2* CKO retina is the relevant control since the underlying genetic context is the same.

– While it could be argued that starting the cultures earlier would make the wild type a more relevant condition to function as a control, the experiment would have to be initiated prior to Shh signaling activation. This is not possible, however, because the culture would have to be initiated prior to E11.5, which is when tamoxifen is administered. Earlier tamoxifen treatment is confounded by the developmental arrest of eye development due to Lhx2 inactivation at a sufficiently earlier stage such as E9.5. This reasoning is further confounded by the issue that the Gas1 and Cdon are expressed at these earlier stages, which again suggests that wild type context is different than the Lhx2 CKO. Based on these considerations, we argue that the experimental design used is the most relevant and interpretable for the question being asked.

– Another consideration is that 3 nM Shh-N was added to the medium in these experiments to compensate for the loss of the Shh producing cells in the suspension culture paradigm (Suppl. Figure 6). This maintains Shh signaling in the wild type retina (Figure 6G), which could mask any potential positive effects of coreceptor overexpression.

– Finally, like many other reporters, β-Galactosidase (βGal) protein is not rapidly downregulated, and this is problematic for conditions where the reporter is already expressed at the start of the experiment, which is the case in the wild type retina. Even if βGal expression dropped below detectable levels within 24 hr of the start of the experiment, its basal level might still be higher than in the *Lhx2* CKO retinal explants, which again complicates the use of the wild type retina for the coreceptor electroporations.

Reviewer 2: Cdon and Gas1 overexpression in the following contexts: wild type, Ptch1CKO, PMA/ShhN exposure, ex vivo culture. Alternatively, an ex vivo or in vitro approach using cultured cells may also prove worthy to prove this point.

To respond to the suggestions made here, we consider each context individually.

– Wild type: This is equivalent to the control retina condition suggested by Reviewer 1 and was discussed above.

– *Ptch1* CKO: *Ptch1* inactivation autonomously activates and stimulates the pathway, which is the same direction observed with coreceptor overexpression. Since the co-receptors act directly upstream of Ptch1 as part of the mechanism to present Shh ligand to Ptch1, it could be difficult to observe additional stimulation of the pathway because of epistasis and/or because the pathway is already overstimulated due to Ptch1 inactivation.

– Purmorphamine exposure: Purmorphamine acts directly on Smoothened to stimulate Shh signaling and purmorphamine stimulation is ligand independent. In contrast, coreceptor mediated stimulation of Shh signaling is ligand dependent and functions upstream of Smoothened (and Ptch1). Purmorphamine would therefore mask any stimulatory effects of the co-receptors on the pathway if combined into a single condition.

– Shh-N exposure: This condition was done in our study. As mentioned above, 3 nM Shh-N was used in the coreceptor electroporation experiments (Figure 8C-E). The coreceptors require ligand to activate the pathway and adding Shh-N to the culture medium was necessary to compensate for the loss of the Shh producing retinal ganglion cells during the culture.

References

Allen, B. L., Song, J. Y., Izzi, L., Althaus, I. W., Kang, J.-S., Charron, F., Krauss, R. S. and McMahon, A. P. (2011). Overlapping roles and collective requirement for the coreceptors GAS1, CDO, and BOC in SHH pathway function. *Dev. Cell* 20, 775–787.

Butterfield, N. C., Metzis, V., McGlinn, E., Bruce, S. J., Wainwright, B. J. and Wicking, C. (2009). Patched 1 is a crucial determinant of asymmetry and digit number in the vertebrate limb. *Development* 136, 3515–3524.

Hägglund, A.-C., Dahl, L. and Carlsson, L. (2011). Lhx2 is required for patterning and expansion of a distinct progenitor cell population committed to eye development. *PLoS ONE* 6, e23387.

Izzi, L., Lévesque, M., Morin, S., Laniel, D., Wilkes, B. C., Mille, F., Krauss, R. S., McMahon, A. P., Allen, B. L. and Charron, F. (2011). Boc and Gas1 each form distinct Shh receptor complexes with Ptch1 and are required for Shh-mediated cell proliferation. *Dev. Cell* 20, 788–801.

Kahn, B. M., Corman, T. S., Lovelace, K., Hong, M., Krauss, R. S. and Epstein, D. J. (2017). Prenatal ethanol exposure in mice phenocopies Cdon mutation by impeding Shh function in the etiology of optic nerve hypoplasia. *Disease Models & Mechanisms* 10, 29–37.

Lee, C. S., Buttitta, L. and Fan, C. M. (2001). Evidence that the WNT-inducible growth arrest-specific gene 1 encodes an antagonist of sonic hedgehog signaling in the somite. *PNAS* 98, 11347–11352.

Li, P., Markson, J. S., Wang, S., Chen, S., Vachharajani, V. and Elowitz, M. B. (2018). Morphogen gradient reconstitution reveals Hedgehog pathway design principles. *Science* 360, 543–548.

Sakagami, K., Gan, L. and Yang, X.-J. (2009). Distinct effects of Hedgehog signaling on neuronal fate specification and cell cycle progression in the embryonic mouse retina. *Journal of Neuroscience* 29, 6932–6944.

Tenzen, T., Allen, B. L., Cole, F., Kang, J.-S., Krauss, R. S. and McMahon, A. P. (2006). The cell surface membrane proteins Cdo and Boc are components and targets of the Hedgehog signaling pathway and feedback network in mice. *Dev. Cell* 10, 647–656.

Wall, D. S., Mears, A. J., McNeill, B., Mazerolle, C., Thurig, S., Wang, Y., Kageyama, R. and Wallace, V. A. (2009). Progenitor cell proliferation in the retina is dependent on Notch-independent Sonic hedgehog/Hes1 activity. *J. Cell Biol.* 184, 101–112.

Wang, Y., Dakubo, G. D., Thurig, S., Mazerolle, C. J. and Wallace, V. A. (2005). Retinal ganglion cell-derived sonic hedgehog locally controls proliferation and the timing of RGC development in the embryonic mouse retina. *Development* 132, 5103–5113.

Wierbowski, B. M., Petrov, K., Aravena, L., Gu, G., Xu, Y. and Salic, A. (2020). Hedgehog Pathway Activation Requires Coreceptor-Catalyzed, Lipid-Dependent Relay of the Sonic Hedgehog Ligand. *Dev. Cell* 55, 450–467.e8.

[Editors' note: further revisions were suggested prior to acceptance, as described below.]

Please also address all the main point raised by the additional reviewer of your manuscript as they will help increasing the clarity of your message. The additional comments points to additional details of the manuscript that need your attention.Reviewer #3:The Lhx2 LIM homeodomain transcription factor plays multiple key roles in vertebrate eye development - specifying regions of the eye field, patterning the optic cup, maintaining the pool of mitotic retinal progenitor cells (RPC), modulating retinal cell fate determination (histogenesis), and integrating signaling pathways. In germline Lhx2 mutants, morphogenesis arrests at the optic vesicle stage.All paracrine signals have two parts - the sending and responding cells. In this study, Li, Gordon et al. explore how neural progenitors acquire the competence to receive and transduce sonic hedgehog signals - via expression of coreceptors and other pathway components, regulated in large part by Lhx2.They deeply explore the mechanism of Lhx2 action in RPCs. They offer plausible explanations for the phenotypic similarity between Lhx2 and Shh retinal CKO (conditional knockout) mice. To test their central hypothesis - that Lhx2 confers RPCs with enhanced competence to respond to Shh signaling by positively regulating expression of Gas1 and Cdon coreceptors - they push and pull the Shh pathway at multiple levels. They employ informative mutants, Hes1-CreERT temporal loss-of-function mice (to delete Lhx2 in all RPCs at the onset of histogenesis, from embryonic day E11.5), pharmacological treatment (N-Shh, PMA agonist), bypass experiments, and creative bioassays (e. g. retinal explant cocultures with apposed open-loop NIH3T3 fluorescent Shh reporter cells). They assess molecular effects of Lhx2 deletion on mRNA (bulk RNA-seq, qPCR) and protein abundance, chromatin accessibility, and informative reporter expression (e. g. Gli1-lacZ knock-in allele) and evaluate Lhx2 genomic targets (ChIP-seq). The various data converge. The authors apply standard pathway logic (molecular epistasis) to define the major steps and general mechanisms through which Lhx2 modulates Shh signaling.The results are complicated and do not fully explain all phenotypes. Lhx2 is neither necessary nor sufficient for Shh action. Signaling is attenuated but not eliminated in mutant RPCs. Despite this complexity, the study advances understanding and provides a basis to explore how signaling pathways are integrated and finely tuned.1. The initial data are presented in a confusing order. After noting similarity between Lhx2 and Shh retinal CKO phenotypes, the authors show in situ hybridization images (Fig 1) suggesting that Shh signaling (Gli1) - but not Shh ligand - is downregulated in Lhx2 CKO retina. This is the major premise for the study - and title of the first Results section. However, the ISH data are not quantitative (or persuasive), and the CKO histological phenotype does not obviously differ from control at E14 ('cell type profiles' in line 132, a bit vague), so the manuscript does not start with a strong foundation. However, the Gli1 and Shh qPCR data (Fig 6B) are convincing and do support ISH findings - and should be presented early, before the Gli1 immunoblot (Fig 1C). Most readers will look for these quantitative mRNA data in Fig 1. They belong here, logically, since Gli1 is a well-established, sensitive readout of Shh signaling.

As suggested, the qPCR data for *Gli1* and *Shh* expression in Fig. 6B is now located in Fig.1G.

We also included an immunohistological comparison of the proliferation markers PCNA and EdU, and Lhx2 from E15.5 *Lhx2* CKO and Ctrl embryos (Fig. 1C). This data shows the persistence of RPCs, lending support to the argument that Gli1 levels are not reduced because of a lack of RPCs at this age. We also provided a more specific description of this histological feature than originally provided per the reviewer’s suggestion.

Additional related changes:

Although the reviewer suggested the qPCR data be provided before the ISH data, we found the logic flow to be difficult and these data are provided after the ISH data, along with qPCR data showing that *Lhx2* mRNA is downregulated within 48 hr of tamoxifen treatment (Fig. 1H), a requested control to establish the timing of *Lhx2* inactivation.

The original Supplemental Fig. 1 in situ hybridizations for *Gli1* and *Shh* at E16.5 following tamoxifen treatment at E12.5 was removed because it was redundant with information provided in Supplemental Fig. 8A. The current Supplemental Fig. 1 contains images showing eye size and marker expression comparisons between the *Lhx2* CKO and controls at E15.5 in support of the new phenotype data provided in Fig. 1C.

Likewise, the Lhx2 RNA-seq data in Fig. 2C-E are confusing. Knowing the structure of the conditional allele in advance, the authors should quantify mRNA isoform reads with WT (ex 1/2, 2/3 or 3/4 junctions), deleted (ex 1/2 junction) or unknown (reads lacking ex 1-3) splicing patterns separately, in their existing dataset. Rather than the simple schema in Fig 1E, they should show 6K WT and CKO reads aligned to the mm10 reference genome, along with the AUG and qPCR amplicon, and note whether deleted isoforms are predicted to undergo NMD and should thus have lower mRNA abundance a priori (in Methods or legend).

We now provide coverage plots (Fig. 2 E) showing the merged read counts for all *Lhx2* CKO and control samples mapped to the Lhx2 locus. We did not map exon junctions to assess isoform combinations, but the data clearly show that all exons except for exons 2 and 3 are expressed at similar levels in the CKO and control, suggesting that NMD is not occurring. This is now mentioned in the legend of Fig. 2.

How much is Lhx2 mRNA downregulated within 48 hr of Tam treatment (line 130)? What are 'other' features in Fig 2B?

Lhx2 mRNA downregulation within 48 hr of tamoxifen treatment is shown by qPCR in Fig. 1H.

Other features are rare transcript biotypes from MGI such as *pseudogenes*, *processed transcripts, antisense RNAs, TEC RNAs.* This is now explained in the results and Fig. 2B with ‘pseudogenes’ as an example.

2. The bioinformatic rationale for exploring Lhx2 regulation of Shh pathway genes (Fig 3) is logical but presented in a jumbled way. Two orthogonal strategies are applied (three, including ATAC-seq) - and only one pathway (Hh) emerged with statistical significance in both. One approach (RNA-seq of dissected CKO retinas, with KEGG pathway topology analysis of DEGs) tests altered mRNA expression - direct and indirect regulation - whereas the ChiP-seq data (from flow-sorted wild-type RPCs in an earlier study, analyzed via clusterProfiler) tests Lhx2 chromatin occupancy - expected for direct regulation. The logic for this overlapping strategy is buried in the text, and obscured by interspersed panels in Fig 3, forcing the reader to carefully dissect the legend to understand what kind of (highly derived) data are presented. Label panels 3AB better so these methods are obvious.

Given the concerns raised here as well as in minor comments 3-6, we did a major revision for this section (Results/Lhx2 inactivation alters the expression of multiple Shh pathway components). We also revised the Methods/Bioinformatics/*RNA-seq data analysis* section, Figure 3, and Figure 3 legend. Listed below are specific issues and changes made:

– The text was rewritten to better clarify the orthogonal comparisons at the levels of pathways and individual genes. The volcano plot was moved to Figure 2.

– More detailed explanations of the underlying statistics for all bioinformatics analyses were moved into the *Bioinformatics* subsection of the Methods.

– Methodological information was moved from the results to the methods.

– We adopted terms to better highlight the meaning of the statistical values in the text and Figure 3A and B. They are defined at first use.

ASP score: Activity State Prediction score - equivalent to the ‘Z-score of Activation’ from CP.

DEG ratio (Fig. 3A; right y-axis): Defined in text, figure, and legend.

Bound gene ratio (Fig. 3B; right y-axis): Defined in text, figure, and legend.

– Panels Fig 3A and B were more comprehensively labeled and incorporated the new terms above. The layout of the panels was modified to provide a more uniform appearance for clarity.

3. The appositional explant assay is a compelling functional measure of bioactive Shh (Fig 5). Likewise, the Lhx2 and Ptch1 CKO and dCKO data (Fig 7) are creative and instructive, showing that Lhx2 acts upstream but that Shh signaling remains intact (attenuated) in Lhx2 CKO RPCs.However, the qPCR normalization (Fig 7A) does not make sense (to me) - each control value should be set to 1.0 +/- SE (or SD). There is a similar normalization concern in Fig 10 (lines 701 ff notwithstanding).

The mean for the control values vary from 1.0 for Figs. 7A and 10 because a single control sample was designated as a reference sample that was repeatedly used on separate runs and all samples were normalized to this reference. We took this approach to account for batch effects that could arise due to samples being run on different plates and/or days, which was necessary to collect the qPCR data in Figs. 7A and 10. This information is now provided in the qPCR method section.

Does the increased abundance of Cyclin D1+ cells (Fig 7) reflect an increase in bioactive Shh mitogen (Fig 5B)?

We believe the reviewer is referring to the increased fluorescence intensities of Cyclin D1+ cells in the mutants rather than the cell population abundance, which is lower in the mutants (Fig. 7C, D). This question is interesting because the increase in Cyclin D1 levels seems counterintuitive to a reduction in proliferation. It could reflect enhanced mitogenic signaling, or it could reflect a cell autonomous change in Cyclin D1 regulation. The implication of this observation could be a compensatory mechanism that sustains proliferation in the absence of Lhx2. Further work would be needed to understand its potential functional significance and mechanism(s) causing this effect.